# Sequence logic at enhancers governs a dual mechanism of endodermal organ fate induction by FOXA pioneer factors

Ryan J. Geusz [1,2,3,4,8], Allen Wang[1,2,3,8], Dieter K. Lam[1,2,3], Nicholas K. Vinckier[1,2,3], Konstantinos-Dionysios Alysandratos[5,6], David A. Roberts[5], Jinzhao Wang [1,2,3], Samy Kefalopoulou[1,2,3], Araceli Ramirez[1,2,3], Yunjiang Qiu [2], Joshua Chiou [1,4], Kyle J. Gaulton [1], Bing Ren [2,7], Darrell N. Kotton [5,6] & Maike Sander [1,2,3✉]

FOXA pioneer transcription factors (TFs) associate with primed enhancers in endodermal organ precursors. Using a human stem cell model of pancreas differentiation, we here discover that only a subset of pancreatic enhancers is FOXA-primed, whereas the majority is unprimed and engages FOXA upon lineage induction. Primed enhancers are enriched for signal-dependent TF motifs and harbor abundant and strong FOXA motifs. Unprimed enhancers harbor fewer, more degenerate FOXA motifs, and FOXA recruitment to unprimed but not primed enhancers requires pancreatic TFs. Strengthening FOXA motifs at an unprimed enhancer near *NKX6.1* renders FOXA recruitment pancreatic TF-independent, induces priming, and broadens the *NKX6.1* expression domain. We make analogous observations about FOXA binding during hepatic and lung development. Our findings suggest a dual role for FOXA in endodermal organ development: first, FOXA facilitates signal-dependent lineage initiation via enhancer priming, and second, FOXA enforces organ cell type-specific gene expression via indirect recruitment by lineage-specific TFs.

[1] Department of Pediatrics, Pediatric Diabetes Research Center, University of California, La Jolla, San Diego, CA 92093, USA. [2] Department of Cellular & Molecular Medicine, University of California, La Jolla, San Diego, CA 92093, USA. [3] Sanford Consortium for Regenerative Medicine, La Jolla, San Diego, CA 92093, USA. [4] Biomedical Graduate Studies Program, University of California San Diego, La Jolla, San Diego, CA 92037, USA. [5] Center for Regenerative Medicine of Boston University and Boston Medical Center, Boston, MA 02118, USA. [6] The Pulmonary Center and Department of Medicine, Boston University School of Medicine, Boston, MA 02118, USA. [7] Ludwig Institute for Cancer Research, La Jolla, San Diego, CA 92093-0653, USA. [8]These authors contributed equally: Ryan J. Geusz, Allen Wang. ✉email: masander@ucsd.edu

The pancreas, liver, and lung develop from the foregut endoderm in response to local signaling cues that specify lineage identity by inducing organ-specific gene expression. The competence of organ lineage precursors to activate lineage-specific genes in response to inductive signals is acquired during endoderm development[1,2]. Coincident with the acquisition of competence, the transcription factors (TFs) FOXA1 and FOXA2 (henceforth abbreviated FOXA1/2) are recruited to enhancers of foregut-derived organ lineages, leading to a gain in chromatin accessibility and H3K4me1 deposition[1,3,4], a phenomenon referred to as enhancer priming. Thus, current evidence suggests that FOXA1/2's role in endodermal organ development is to render foregut endoderm competent to activate organ-specific genes by broadly priming pancreas-, liver-, and lung-specific enhancers before organ-inductive signals trigger enhancer activation. Consistent with this model, studies in model organisms and human pluripotent stem cell (hPSC)-based differentiation systems have shown a requirement for FOXA1/2 in pancreas, liver, and lung development, with the two FOXA TFs functioning in a partially or fully redundant manner[3–6]. However, whether chromatin priming is the only mechanism by which FOXA TFs control endodermal organ development is unknown.

The mechanisms by which FOXA TFs engage with and open chromatin have been the subject of debate. In vitro experiments have shown that FOXA TFs possess pioneering activity, which refers to the specific ability of a TF to engage target sites on nucleosomal DNA and to remodel such regions to increase chromatin accessibility[7–9]. Through their chromatin remodeling activity, FOXA TFs facilitate subsequent binding of other TFs and co-factors that further modify chromatin state and initiate gene expression[9–14]. However, despite their ability to access target sites in closed chromatin in vitro, binding site selection of FOXA and other pioneer TFs in cellular contexts has been shown to depend on additional features, such as the local chromatin landscape[15], presence of cooperative binding partners[16,17], and strength of the binding motif[17–19]. For example, steroid receptor activation in breast cancer cell lines induces FOXA1 recruitment to sites with degenerate FOXA1-binding motifs[18,20], exemplifying heterogeneity in FOXA target site engagement. The determinants that underlie FOXA-binding site selection and FOXA-mediated enhancer priming during cellular transitions of development remain to be explored.

Here, we sought to determine the specific mechanisms that underlie the regulation of endodermal organ development by FOXA TFs. To this end, we mapped FOXA1/2 genomic association with pancreas-specific enhancers throughout a time course of hPSC differentiation into pancreas. Surprisingly, only a minority of pancreas-specific enhancers are FOXA1/2-bound prior to lineage induction and exhibit priming, whereas the majority engage FOXA1/2 concomitant with pancreas induction. Compared to unprimed enhancers, primed enhancers contain DNA sequences more closely matching FOXA consensus motifs and harbor additional sequence motifs for signal-dependent TFs. By contrast, unprimed enhancers contain degenerate and fewer FOXA motifs, are enriched for motifs of lineage-specific TFs, and depend on the pancreas-specific TF PDX1 for FOXA1/2 recruitment. We further show that CRISPR/Cas9-mediated optimization of FOXA motifs in an unprimed enhancer near the pancreatic TF NKX6.1 is sufficient to redefine patterns of FOXA binding and to broaden NKX6.1 expression within the pancreatic progenitor domain, suggesting that FOXA motif strength is relevant for fine-tuning developmental gene expression. In-depth analysis of FOXA binding during hPSC differentiation toward hepatocytes and lung alveolospheres revealed similar patterns of FOXA binding and sequence logic at FOXA-bound enhancers. Our findings show that FOXA1/2 regulate foregut organ development through two distinct and complementary mechanisms: priming of a small subset of organ-specific enhancers before lineage induction and activation of a larger cohort of enhancers through cooperative binding with organ lineage-specific TFs. We propose that priming of a small enhancer subset permits precise spatial and temporal regulation of organ induction by lineage-inductive signals, whereas cooperative FOXA binding with lineage-specific TFs ensures cell type specificity of gene expression, providing a safeguard against broad activation of alternative lineage programs during developmental transitions.

## Results

**FOXA1 and FOXA2 are necessary for pancreatic lineage induction.** To investigate the role of FOXA1/2 in pancreas development, we employed a hPSC differentiation protocol in which cells transition stepwise to the pancreatic fate through sequential exposure to developmental signaling cues (Fig. 1a). The pancreatic lineage is induced by retinoic acid from gut tube (GT) intermediates, resulting in expression of the pancreatic markers PDX1 in early pancreatic progenitors (PP1) and NKX6.1 in late pancreatic progenitors (PP2). FOXA1 and FOXA2 were expressed from the definitive endoderm (DE) stage onwards (Supplementary Fig. 1a, b), and levels of FOXA1 and FOXA2 were similar in GT, PP1, and PP2 (Supplementary Fig. 1a).

To determine a possible requirement for FOXA1 and FOXA2 in pancreas development, we deleted FOXA1 or FOXA2 in human embryonic stem cells (hESCs) (Fig. 1a and Supplementary Fig. 1c, d) and differentiated control, FOXA1−/−, and FOXA2−/− hESC lines into pancreatic progenitors. Analysis of PDX1 and NKX6.1 expression revealed a requirement for FOXA2 but not FOXA1 for pancreatic lineage induction (Fig. 1b and Supplementary Fig. 2), consistent with recent findings[3]. The presence of residual PDX1+ and NKX6.1+ cells and increased FOXA1 levels in FOXA2−/− pancreatic progenitors (Fig. 1b, c) suggests FOXA1 partially compensates for FOXA2 deficiency. Therefore, we generated FOXA1−/−;FOXA2−/− (FOXA1/2−/−) hESC lines (Supplementary Fig. 1e) and analyzed phenotypes at the DE, GT, and PP2 stages. At the DE and GT stages, similar numbers of FOXA1/2−/− and control cells expressed the DE marker SOX17 and GT marker HNF1B, respectively (Supplementary Fig. 1f, g). In contrast, pancreas induction was blocked in FOXA1/2−/− cells, as evidenced by an almost complete absence of PDX1+ and NKX6.1+ cells, reduced expression of early pancreatic TFs, and down-regulation (≥2-fold change, FDR ≤ 0.05) of genes associated with pancreas-specific biological processes (Fig. 1d–f and Supplementary Data 1, 2). Principal component analysis (PCA) of transcriptome data further confirmed that FOXA1/2−/− and control cells were more similar at the GT stage than at the PP2 stage (Fig. 1g). Together, these findings show that FOXA1 and FOXA2 control pancreatic lineage induction from gut tube lineage intermediates in a partially redundant manner.

**FOXA transcription factors exhibit two temporal patterns of recruitment to pancreatic enhancers.** To identify transcriptional targets of FOXA1/2 during pancreatic lineage induction, we mapped FOXA1/2-binding sites at the GT and PP2 stages. Consistent with the partial functional redundancy between FOXA1 and FOXA2 (Fig. 1b–d), FOXA1 and FOXA2-binding sites were highly correlated at both stages (Supplementary Fig. 3a). FOXA1/2 mostly bound to distal sites (> 2.5 kb from TSS; Supplementary Fig. 3b), suggesting regulation of enhancers by FOXA1/2. To test this, we defined GT and PP2 enhancers as distal H3K27ac peaks (> 2.5 kb from TSS) and compared enhancer activity based on H3K27ac signal in control and FOXA1/2−/− cells at the GT and the PP2 stages. Like gene expression (Fig. 1g), H3K27ac profiles in FOXA1/2−/− and

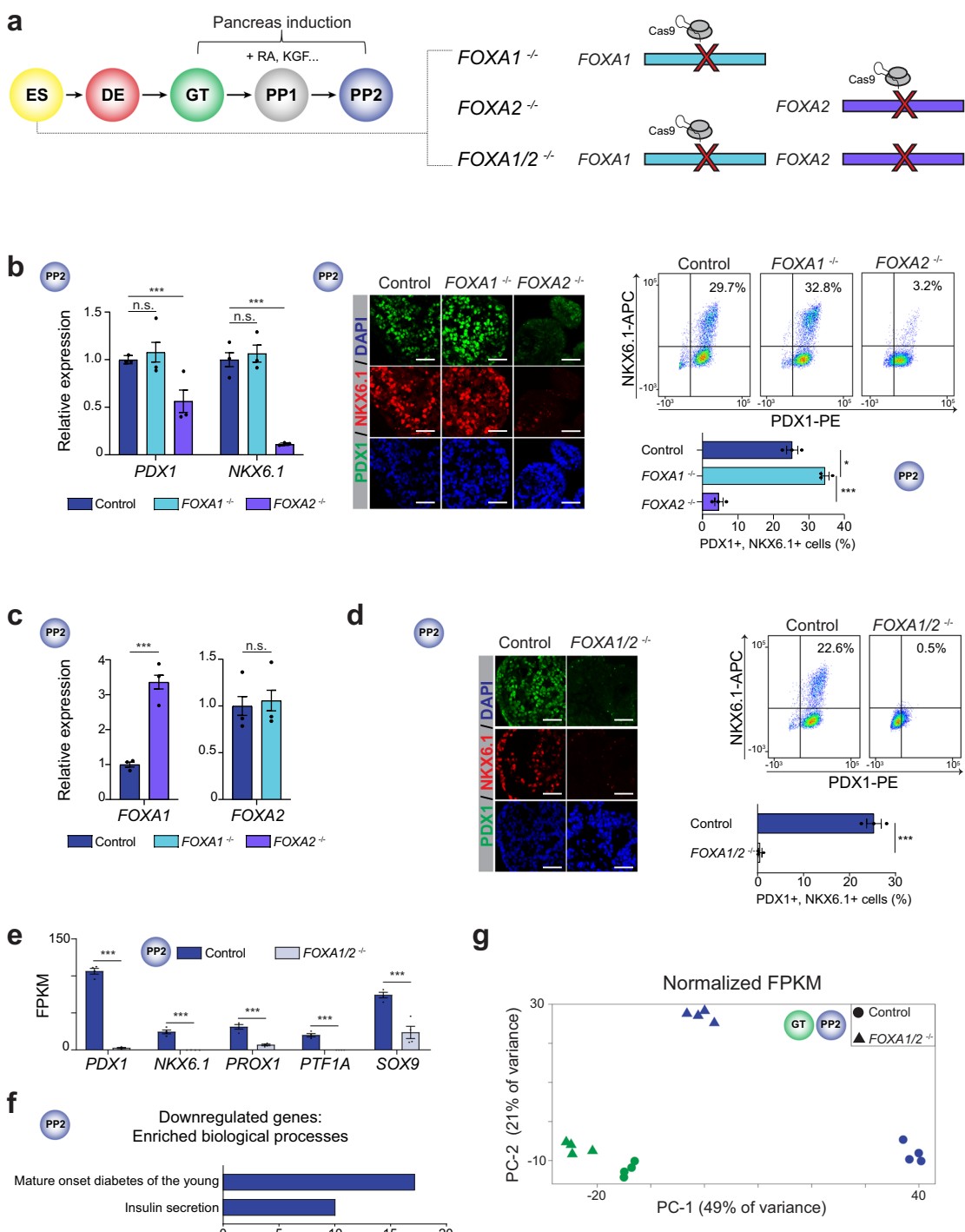

control cells differed more substantially at the PP2 than at the GT stage (Supplementary Fig. 3c), showing that *FOXA1/2* deletion has broad impact on regulation of enhancer activity during the GT to PP2 transition.

To investigate specific mechanisms by which FOXA1/2 mediates pancreatic lineage induction, we identified all FOXA1/2-bound pancreatic enhancers that are activated upon pancreatic lineage induction. To this end, we first identified enhancers that exhibited a ≥ 2-fold increase in H3K27ac signal from the GT to the PP2 stage (2574 enhancers, hereafter referred to as pancreatic enhancers; Supplementary Fig. 3d, e). As expected, genes near these enhancers were predicted to regulate biological processes

associated with pancreas development. Second, we analyzed FOXA1/2 binding at these pancreatic enhancers, revealing that 72% were FOXA1/2-bound at the PP2 stage (Supplementary Fig. 3f). Consistent with prior reports[1,3], we observed FOXA1/2 occupancy at the GT stage preceding pancreatic lineage induction. Surprisingly, however, the percentage of pancreatic enhancers bound by FOXA1/2 was significantly lower at the GT compared to the PP2 stage, implying that not all pancreatic enhancers engage FOXA1/2 before lineage induction. To comprehensively characterize temporal patterns of FOXA1/2 recruitment, we identified all pancreatic enhancers with FOXA1 or FOXA2 binding at the GT and/or PP2 stages and quantified

**Fig. 1 Partially redundant requirement for FOXA1 and FOXA2 in pancreatic lineage induction. a** Schematic of stepwise pancreatic differentiation protocol from hESCs (ES): definitive endoderm (DE), primitive gut tube (GT), early pancreatic progenitor cells (PP1), and late pancreatic progenitor cells (PP2), with indicated genetic modifications in ES. RA, retinoic acid; KGF, keratinocyte growth factor. **b** qPCR analysis of *PDX1* and *NKX6.1* (left), immunofluorescent staining (middle), and flow cytometry quantification of PDX1[+] and NKX6.1[+] cells (right) in control, *FOXA1[−/−]* and *FOXA2[−/−]* PP2 cells (qPCR: $P = 0.493$, 0.590, $3.12 \times 10^{-3}$, and $<1.00 \times 10^{-6}$ for *PDX1* and *NKX6.1* in control compared to *FOXA1[−/−]* and *FOXA2[−/−]* PP2 cells, respectively; flow cytometry: $P = 1.15 \times 10^{-2}$ and $7.00 \times 10^{-4}$ in control compared to *FOXA1[−/−]* and *FOXA2[−/−]* PP2 cells, respectively). **c** qPCR analysis of *FOXA1* and *FOXA2* in control, *FOXA1[−/−]* and *FOXA2[−/−]* PP2 cells ($P = <1.00 \times 10^{-6}$ and 0.700 for *FOXA1* and *FOXA2* in control compared to *FOXA1[−/−]* and *FOXA2[−/−]* PP2 cells, respectively). **d** Immunofluorescent staining (left) and flow cytometry quantification (right) of PDX1[+] and NKX6.1[+] cells in control and *FOXA1/2[−/−]* PP2 cells. ($P = 2.6 \times 10^{-3}$ in control compared to *FOXA1/2[−/−]* PP2 cells). **e** mRNA expression levels of pancreatic transcription factors determined by RNA-seq in control and *FOXA1/2[−/−]* PP2 cells ($n = 4$ independent differentiations; $P$ adj. $= 1.08 \times 10^{-42}$, $2.56 \times 10^{-12}$, $4.93 \times 10^{-20}$, $1.00 \times 10^{-49}$, and $2.82 \times 10^{-4}$ for *PDX1*, *NKX6.1*, *PROX1*, *PTF1A*, and *SOX9*, respectively; DESeq2; FPKM, fragments per kilobase per million fragments mapped). **f** Enriched gene ontology terms of 2833 downregulated genes ($\geq$2-fold decrease, $P$ adj. < 0.05) in *FOXA1/2[−/−]* compared to control PP2 cells. **g** Principal component analysis showing variance in total normalized transcriptome between control and *FOXA1/2[−/−]* cells in GT and PP2. Each point represents one biological replicate. For all qPCR and flow cytometry experiments, $n = 3$ independent differentiations with significance calculated using a 2-sided student's *t*-test. For all immunofluorescence, representative images are shown from $n \geq 2$ independent differentiations; scale bars, 50 μm. All bar graphs show mean ± S.E.M.

FOXA1/2 ChIP-seq signal at these sites (Fig. 2a). We observed three distinct patterns of FOXA1/2 occupancy: class I enhancers (561) were bound by FOXA1/2 at both the GT and PP2 stages, class II enhancers (1422) were FOXA1/2-bound only at the PP2 stage, and the overall small group of class III enhancers (118) was FOXA1/2-bound only at the GT stage (Fig. 2a and Supplementary Data 3–5). Analysis of H3K27ac signal intensity at the GT and PP2 stages showed similar patterns of H3K27ac signal at all enhancers (Fig. 2b), suggesting that enhancers of all classes are mostly inactive at the GT stage and become activated during pancreatic lineage induction. Activation of enhancers of all classes during the GT to PP2 transition was dependent on FOXA1/2 (Fig. 2c and Supplementary Fig. 3g). Since the predominant patterns were either maintenance of FOXA1/2 binding (class I) or de novo FOXA1/2 occupancy (class II) after pancreas induction, we excluded class III enhancers from further analyses. We identified examples of both class I and class II enhancers in proximity to gene bodies of pancreatic lineage-determining TFs, such as *PDX1*, *HNF1B*, *NKX6.1*, and *MNX1* (Fig. 2d). Consistent with the H3K27ac pattern, the *PDX1* class I enhancer and the *NKX6.1* class II enhancer are both inactive in GT and active in PP2 in enhancer reporter assays[1]. Together, this analysis shows that FOXA1/2 recruitment to pancreatic enhancers precedes lineage induction at only a small subset of enhancers, while FOXA1/2 recruitment to most pancreatic enhancers coincides with lineage induction (Fig. 2e).

**Primed and unprimed pancreatic enhancers reside in distinct regulatory domains**. Given early recruitment of FOXA1/2 to class I but not class II enhancers, we hypothesized that the two classes could differ in their temporal pattern of gain in chromatin accessibility and H3K4me1 deposition, predicting that early FOXA1/2 occupancy at class I enhancers would lead to chromatin priming. As predicted, class I enhancers exhibited open chromatin and H3K4me1 deposition at the GT stage (Fig. 3a and Supplementary Fig. 4a, b). By contrast, class II enhancers acquired these features largely with pancreatic lineage induction (Fig. 3a and Supplementary Fig. 4a, b), identifying primed chromatin as a feature of class I enhancers. Although a subset of class II enhancers was marked by H3K4me1 at the GT stage, this population comprised the minority of class II enhancers (Supplementary Fig. 4c). At both class I and class II enhancers, H3K4me1 deposition and gain in chromatin accessibility during lineage induction was FOXA1/2-dependent (Fig. 3b and Supplementary Fig. 4b), demonstrating that FOXA1/2 are necessary for chromatin remodeling at both classes of enhancers.

We next sought to determine whether class I and class II enhancers function together within larger regions of active

chromatin such as super-enhancers[21], or whether they reside in distinct regulatory domains. To distinguish between these possibilities, we defined 167 super-enhancers among the 2574 pancreatic enhancers identified in Supplementary Fig. 3d (Supplementary Fig. 4d and Supplementary Data 6) and found that 160 (96%) were FOXA1/2-bound at the PP2 stage (Supplementary Fig. 4e). Analysis of overlap between class I or class II enhancers and FOXA-bound super-enhancers revealed that most FOXA-bound super-enhancers (76%) contained either class I or class II enhancers but not both (Fig. 3c). Furthermore, we analyzed Hi-C datasets produced from PP2 stage cells and found that class I and class II enhancers were mostly located in non-overlapping 3D chromatin loops (Fig. 3d and Supplementary Data 7). This evidence indicates that class I and class II enhancers reside largely within distinct gene regulatory domains and therefore likely function independently.

To identify target genes of class I and class II enhancers, we assigned enhancers to their nearest expressed gene at the PP2 stage (Supplementary Data 8,9), and validated predictions by showing regulation of these genes by FOXA1/2 (Supplementary Fig. 4f). Consistent with their location in distinct regulatory domains (Fig. 3c, d), class I and class II enhancers mostly associated with distinct genes, including pancreatic lineage-determining TFs (Fig. 3e). Of note, gene ontology analysis of genes regulated by class I compared to class II enhancers revealed roles for class I enhancer-associated genes in cellular signal transduction pathways (Supplementary Fig. 4g and Supplementary Data 10), whereas no comparative enrichment of specific gene ontology terms was observed for class II enhancer-associated genes. Together, these results suggest that two distinct mechanisms establish the pancreatic gene expression program: a subset of pancreatic genes is regulated by enhancers that undergo FOXA1/2-mediated chromatin priming at the gut tube stage, whereas most pancreatic genes are regulated by enhancers that are unprimed prior to pancreatic lineage induction, and to which FOXA1/2 are recruited upon lineage induction (Fig. 3f).

**Distinct DNA sequence motifs at primed and unprimed pancreatic enhancers**. We next investigated mechanisms that could explain the observed temporal differences in FOXA1/2 binding to class I (primed) and class II (unprimed) pancreatic enhancers. To test whether differences in DNA sequence could provide an explanation, we conducted de novo motif analysis to identify motifs enriched at class I enhancers against a background of class II enhancers. Class I enhancers were enriched for FOXA motifs and motifs for several signal-dependent TFs, including the ETS family TFs GABPA and SPDEF, the downstream effector of Hippo signaling TEAD, and the retinoic acid receptor RXRA

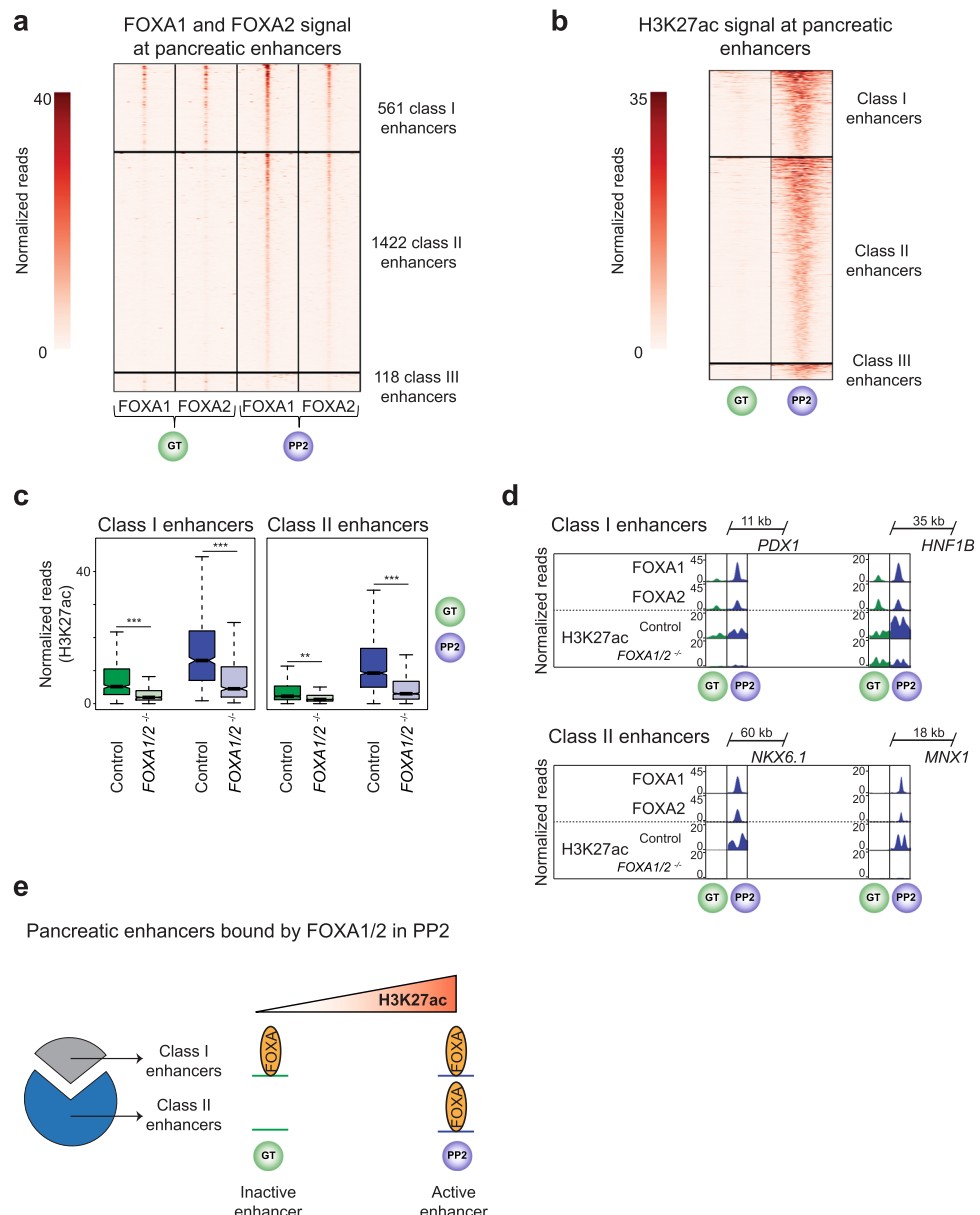

**Fig. 2 Two distinct temporal patterns of FOXA1 and FOXA2 binding to pancreatic enhancers. a** and **b** Heatmaps showing density of FOXA1 and FOXA2 ChIP-seq reads (**a**) and H3K27ac ChIP-seq reads (**b**) at pancreatic enhancers in GT and PP2. Heatmaps are centered on FOXA1, FOXA2, and H3K27ac peaks, respectively, and span 5 kb. Pancreatic enhancers are classified based on temporal pattern of FOXA1 and FOXA2 occupancy. **c** Box plots of H3K27ac ChIP-seq counts at class I and class II pancreatic enhancers in control and *FOXA1/2*−/− GT and PP2 cells ($P = < 2.2 \times 10^{-16}$, $<2.2 \times 10^{-16}$, 0.009, and $<2.2 \times 10^{-16}$ for control versus *FOXA1/2*−/− at class I enhancers in GT, class I enhancers in PP2, class II enhancers in GT, and class II enhancers in PP2, respectively; Wilcoxon rank sum test, 2-sided). Plots are centered on median, with box encompassing 25th–75th percentile and whiskers extending up to 1.5 interquartile range. **d** Genome browser snapshots showing FOXA1, FOXA2, and H3K27ac ChIP-seq signal at class I pancreatic enhancers near PDX1 and HNF1B and class II pancreatic enhancers near NKX6.1 and MNX1 in GT and PP2. Approximate distance between enhancer and gene body is indicated. **e** Schematic illustrating the identified pattern of FOXA1/2 occupancy at pancreatic enhancers. All ChIP-seq experiments, $n = 2$ replicates from independent differentiations.

(Fig. 4a and Supplementary Data 11). Work in model organisms has identified critical roles for ETS TFs as well as Hippo and retinoic acid signaling in early pancreatic development[22–25], suggesting that pancreatic lineage-inductive signals are read at class I enhancers by partnering of FOXA1/2 with signal-dependent TFs. ChIP-seq analysis for RXR confirmed preferential RXR binding to class I compared to class II enhancers at the PP1 stage (Fig. 4b). Class I enhancers were also enriched for GATA TF motifs (Fig. 4a), and a higher percentage of class I than class II enhancers bound GATA4 and GATA6 at the GT stage

(Fig. 4b). Given that GATA TFs cooperatively bind with FOXA1/2 to DNA[17], GATA4/6 could facilitate FOXA1/2 recruitment to a subset of class I enhancers prior to pancreas induction.

Since FOXA1/2 binding to class I enhancers precedes binding to class II enhancers (Fig. 2a) and FOXA motifs are enriched at class I compared to class II enhancers (Fig. 4a), we postulated that different mechanisms could underlie FOXA1/2 recruitment to the two classes of enhancers. Binding site selection of pioneer TFs such as FOXA1/2 has been shown to depend on motif abundance, strength, and position[17–19,26]. Therefore, we analyzed FOXA

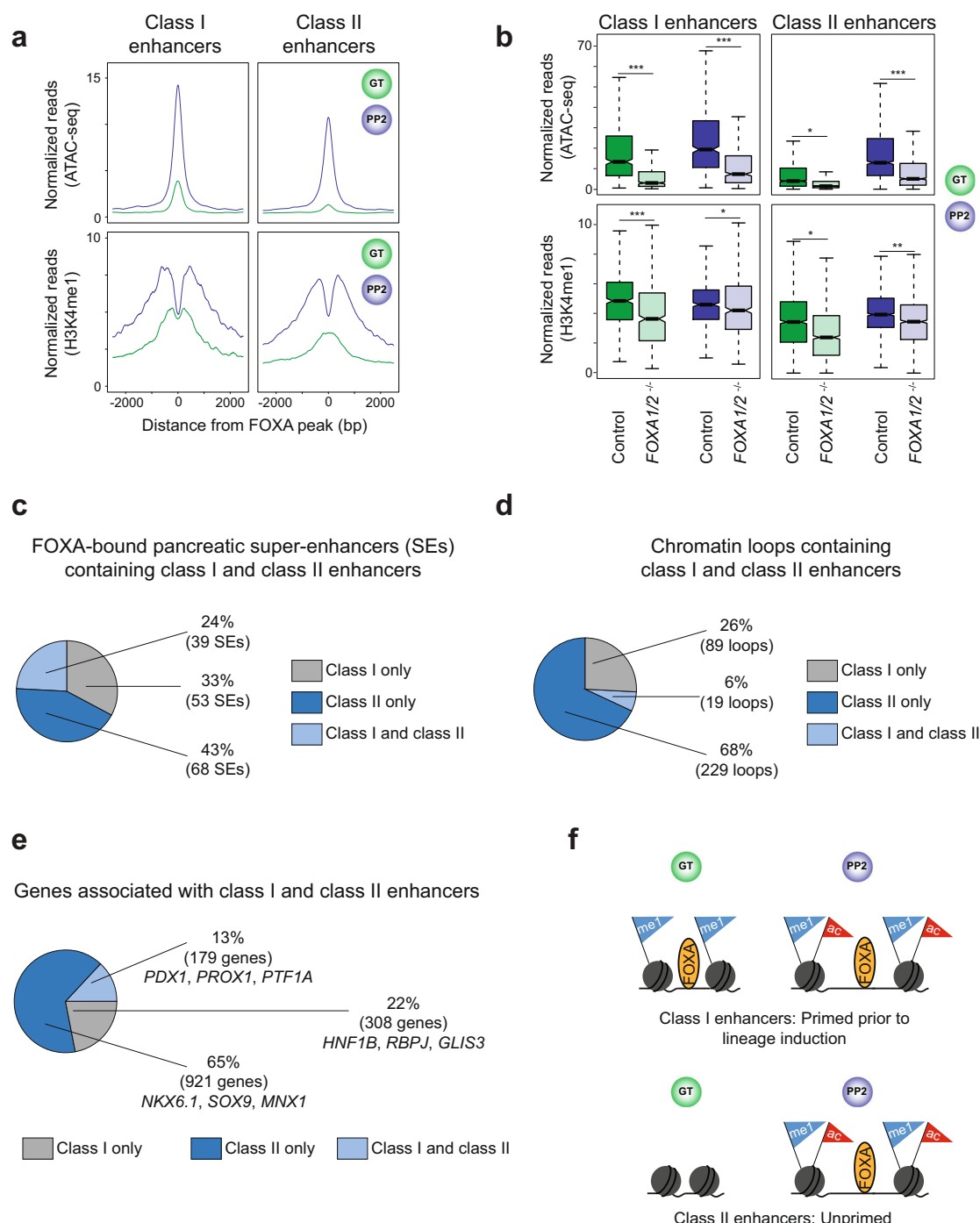

**Fig. 3 Class I and class II pancreatic enhancers largely map to distinct gene regulatory elements. a** Tag density plots for class I and class II pancreatic enhancers displaying ATAC-seq (top) and H3K4me1 ChIP-seq (bottom) read density in GT and PP2. Plots are centered on FOXA1/2 peaks and span 5 kb. **b** Box plots of ATAC-seq (top) and H3K4me1 ChIP-seq (bottom) counts at class I and class II pancreatic enhancers in GT and PP2 for control and *FOXA1/2*$^{-/-}$ cells ($P = < 2.2 \times 10^{-16}$, $<2.2 \times 10^{-16}$, 0.01, and $<2.2 \times 10^{-16}$ for control versus *FOXA1/2*$^{-/-}$ of ATAC-seq signal at class I in GT, class I in PP2, class II in GT, and class II in PP2, respectively. $P = < 2.2 \times 10^{-16}$, 0.01, 0.02, and 0.01 for control versus *FOXA1/2*$^{-/-}$ of H3K4me1 signal at class I in GT, class I in PP2, class II in GT, and class II in PP2, respectively; Wilcoxon rank sum test, 2-sided). Plots are centered on median, with box encompassing 25th–75th percentile and whiskers extending up to 1.5 interquartile range. **c** Percentage of FOXA1- and/or FOXA2-bound pancreatic super-enhancers (SEs) in PP2 containing only class I, only class II, or both class I and class II enhancers. **d** Percentage of chromatin loop anchors in PP2 containing only class I, only class II, or both class I and class II enhancers. **e** Percentage of genes associated with only class I, only class II, or both class I and class II enhancers. Target genes were assigned to enhancers based on nearest TSS of expressed genes (fragments per kilobase per million fragments mapped (FPKM) ≥ 1) in PP2. **f** Schematic illustrating FOXA1/2 occupancy, chromatin accessibility, and presence of H3K4me1 and H3K27ac at class I and class II enhancers in GT and PP2. All ChIP-seq and ATAC-seq experiments, $n = 2$ replicates from independent differentiations.

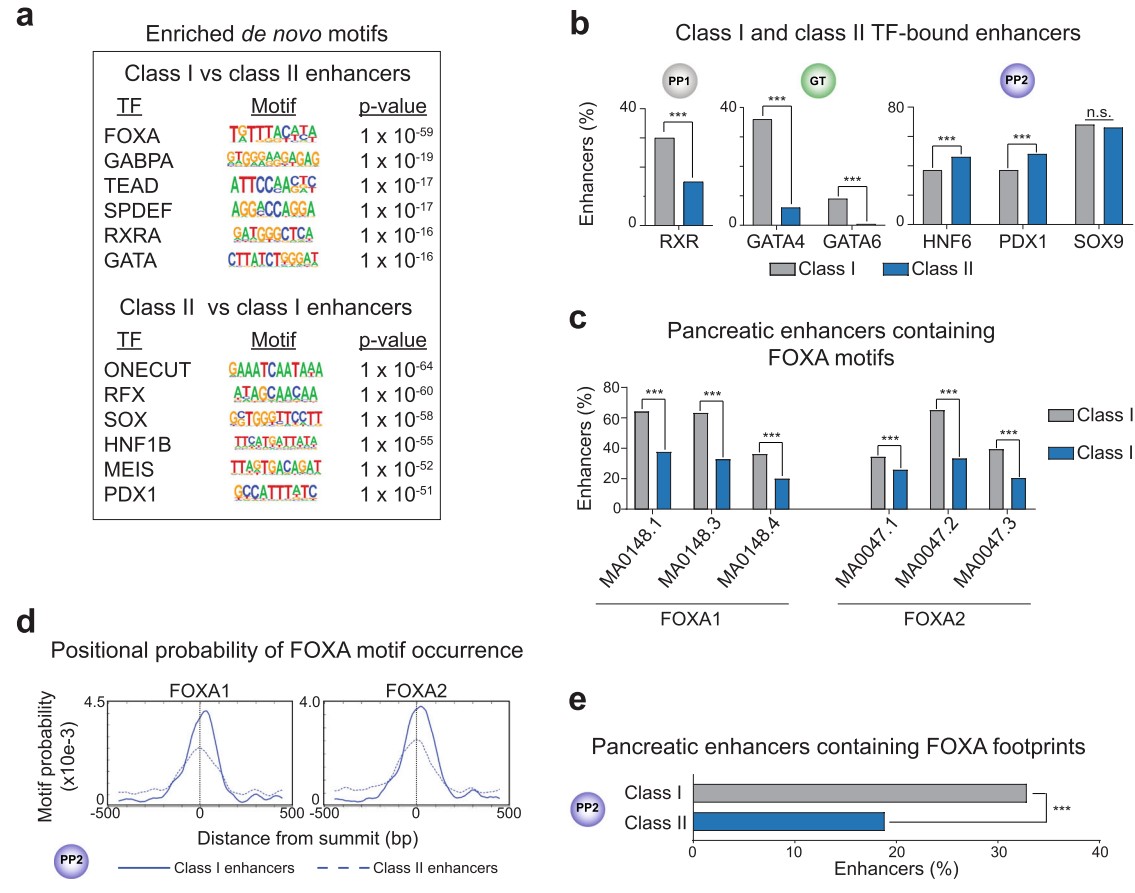

**Fig. 4 FOXA1/2-binding sites at class I and class II pancreatic enhancers differ in DNA sequence. a** Enriched de novo transcription factor (TF)-binding motifs at class I against a background of class II pancreatic enhancers and vice versa. Fisher's exact test, 1-sided, corrected for multiple comparisons. **b** Percentage of class I and class II enhancers overlapping RXR ChIP-seq peaks in PP1; GATA4 and GATA6 ChIP-seq peaks in GT; and HNF6, PDX1, and SOX9 ChIP-seq peaks (within 100 bp from peak) in PP2 ($P = 8.27 \times 10^{-14}$, $<2.2 \times 10^{-16}$, $<2.2 \times 10^{-16}$, $3.52 \times 10^{-4}$, $1.01 \times 10^{-5}$, and 0.40 for comparisons of overlap with binding sites for RXR, GATA4, GATA6, HNF6, PDX1, and SOX9, respectively; Fisher's exact test, 2-sided). **c** Percentage of class I and class II enhancers with at least one occurrence of selected FOXA1 and FOXA2 motifs ($P = < 2.2 \times 10^{-16}$, $<2.2 \times 10^{-16}$, $1.76 \times 10^{-13}$, $1.61 \times 10^{-4}$, $<2.2 \times 10^{-16}$, and $<2.2 \times 10^{-16}$ for comparisons of occurrences of MA0148.1, MA0148.3, MA0148.4, MA0047.1, MA0047.2, and MA0047.3, respectively. Fisher's exact test, 2-sided). **d** Probability (motif occurrence per base pair) of FOXA1 (MA0148.3) and FOXA2 (MA0047.2) motifs relative to ATAC-seq peak summits at class I (solid line) and class II (dashed line) enhancers. ATAC-seq peak summits at class I enhancers are enriched for occurrences of MA0148.3 ($P = 2.1 \times 10^{-14}$; Fisher's exact test, 1-sided) and MA0047.2 ($P = 6.8 \times 10^{-14}$) compared to summits at class II enhancers. **e** Percentage of class I and class II enhancers containing FOXA TF ATAC-seq footprints in PP2 ($P = 1.01 \times 10^{-10}$ for comparison of class I and class II enhancers; Fisher's exact test, 2-sided). All ChIP-seq experiments, $n = 2$ replicates from independent differentiations.

motifs at class I and class II enhancers for these features. To determine abundance and strength of FOXA motifs, we selected position-weighted matrices (PWMs) corresponding to three FOXA1 and three FOXA2 motifs from JASPAR[27] (Supplementary Fig. 5a), identified occurrences of each motif at class I and class II enhancers, and generated a log-odds score to measure how closely the DNA sequence at each identified motif occurrence matched the PWM. Class I enhancers were significantly enriched for occurrences of all six FOXA motifs compared to class II enhancers (Fig. 4c). Furthermore, three of the FOXA motifs had significantly higher log-odds scores at class I than class II enhancer occurrences (MA0047.2, MA0148.1, and MA0148.3; $P = 1.54 \times 10^{-2}$, $1.10 \times 10^{-3}$, and $1.03 \times 10^{-2}$, respectively; Wilcoxon rank sum test). Thus, class II enhancers contain more degenerate and fewer FOXA motifs compared to class I enhancers. We additionally examined the positioning of FOXA motifs relative to open chromatin by identifying regions of greatest chromatin accessibility at class I and class II enhancers in PP2 stage cells ($n = 531$ and $n = 1257$ ATAC-seq summits in class I and class II enhancers, respectively) and determining

enrichment of each FOXA motif at these regions. Occurrence of all FOXA motifs was enriched at ATAC-seq summits at class I compared to class II enhancers (Fig. 4d and Supplementary Fig. 5b), indicating that regions of greatest chromatin accessibility at class I enhancers are more likely to harbor FOXA motifs. ATAC-seq footprinting analysis further revealed a higher occurrence of FOXA footprints at class I than at class II enhancers (Fig. 4e), indicative of either longer FOXA1/2 DNA residence times or more direct interaction of FOXA1/2 with DNA at class I enhancers[28]. Together, this analysis reveals features of FOXA motifs at class I pancreatic enhancers previously associated with canonical FOXA1/2 pioneer TF activity[17,18].

To further elucidate differences in mechanisms of FOXA recruitment to class I and class II enhancers, we identified de novo motifs enriched at class II enhancers against a background of class I enhancers. Here, we observed enrichment of motifs for pancreatic lineage-determining TFs, such as ONECUT (HNF6), SOX (SOX9), HNF1B, and PDX1 (Fig. 4a and Supplementary Data 12), which sharply increased in expression during pancreatic lineage induction (Supplementary Fig. 5c). To determine whether

these TFs exhibit preferential binding to class II enhancers, we mapped HNF6, PDX1, and SOX9-binding sites genome-wide at the PP2 stage (Fig. 4b and Supplementary Fig. 5d). Overall, we found that similar percentages of class I and class II enhancers were bound by HNF6, PDX1, and SOX9 at the PP2 stage (Fig. 4b). To determine whether the difference in sequence motif enrichment between class I and class II enhancers is also observed when focusing on enhancers bound by a specific TF, we analyzed motifs at HNF6-, PDX1-, or SOX9-bound enhancers. Still, class I enhancers were enriched for FOXA and class II enhancers for ONECUT (HNF6), PDX1, and SOX motifs (Supplementary Fig. 5e and Supplementary Data 13-18). Thus, despite differences in DNA sequence motifs between primed (class I) and unprimed (class II) enhancers, both classes of enhancers are occupied by FOXA1/2, as well as pancreatic lineage-determining TFs after pancreatic lineage induction.

**FOXA1/2 binding to a subset of unprimed enhancers depends on PDX1.** Since motifs for pancreatic lineage-determining TFs, such as PDX1, were enriched at class II compared to class I enhancers (Fig. 4a), we hypothesized that FOXA1/2 recruitment to class II enhancers could require cooperativity with lineage-determining TFs. To test this, we analyzed FOXA1/2 binding, chromatin accessibility, and H3K27ac signal in *PDX1*-deficient pancreatic progenitors (Fig. 5a and Supplementary Fig. 6a). Focusing on PDX1-bound enhancers (n = 205 class I enhancers and 682 class II enhancers), we found that loss of *PDX1* reduced FOXA1/2 binding to a greater extent at class II than class I enhancers (Fig. 5b and Supplementary Fig. 6b, c), exemplified by class I enhancers near *PDX1* and *HNF1B*, and class II enhancers near *NKX6.1* and *MNX1* (Fig. 5c). In total, 23% of PDX1-bound class II enhancers exhibited a significant loss (≥ 2-fold decrease, *P*. adj. < 0.05) in FOXA1/2 ChIP-seq signal after *PDX1* knockdown compared to only 3% of PDX1-bound class I enhancers (Supplementary Fig. 6b). Furthermore, PDX1-bound class II enhancers showed greater loss of FOXA1/2 signal than PDX1-bound class I enhancers (Supplementary Fig. 6c). Given substantial overlap between binding sites for pancreatic lineage-determining TFs (Supplementary Fig. 5d), it is possible that other TFs recruit FOXA1/2 to PDX1-bound class II enhancers where FOXA1/2 occupancy is not significantly affected. Loss of *PDX1* led to a significant reduction in ATAC-seq and H3K27ac signal at both class I and class II enhancers (Supplementary Fig. 6d), showing that full acquisition of chromatin accessibility and enhancer activation during pancreas induction require PDX1 at primed and unprimed enhancers.

Collectively, our findings show that despite similar mechanisms for their activation, primed and unprimed pancreatic enhancers differ in sequence logic and mechanism of FOXA1/2 recruitment (Fig. 5d). Primed enhancers have abundant and strong FOXA motifs, and FOXA1/2 are recruited to primed enhancers prior to pancreatic lineage induction largely independent of the pancreatic TF PDX1. By contrast, unprimed enhancers have fewer and weaker FOXA motifs, and a proportion of unprimed enhancers requires PDX1 for FOXA1/2 recruitment.

**Altering FOXA motif strength redefines temporal FOXA1/2-binding patterns.** We next sought to determine the extent to which the timing and mechanism of FOXA1/2 recruitment are solely dependent on DNA sequence. Since stronger FOXA motifs are a characteristic of class I enhancers, we tested this by optimizing FOXA motifs at a class II enhancer via CRISPR-Cas9 genome editing and mapping FOXA1/2 binding. For this we selected an unprimed class II enhancer near *NKX6.1* for editing in hESCs. This enhancer lacks FOXA1/2 binding (Fig. 2d),

accessible chromatin (Supplementary Fig. 4b), and H3K4me1-signal (Supplementary Fig. 4b) prior to pancreas induction. Furthermore, in the absence of PDX1, FOXA1/2 do not bind to this enhancer (Fig. 5c). Examination of the *NKX6.1* enhancer revealed four degenerate FOXA motifs surrounding the ATAC-seq summit (Fig. 6a). We altered six base pairs within the enhancer to strengthen the FOXA motifs (referred to as motif optimized) (Fig. 6a). Optimizing FOXA motifs resulted in FOXA1/2 recruitment to the *NKX6.1* enhancer at the GT stage prior to pancreas induction (Fig. 6b). Early FOXA1/2 recruitment was accompanied by H3K4me1 but not H3K27ac deposition at the GT stage (Fig. 6b), supporting that FOXA1/2 prime enhancers prior to activation. Thus, optimization of FOXA-binding motifs is sufficient to convert an unprimed class II enhancer into a primed class I enhancer.

**Optimizing FOXA motifs broadens the domain of target gene expression.** To define the relationship between FOXA motif strength and *NKX6.1* target gene expression, we conducted single-cell RNA-sequencing of PP2 cells from control and motif optimized cell lines. Consistent with prior studies[29], we observed a population of multipotent pancreatic progenitor cells expressing high levels of pancreatic lineage-determining TFs (e.g., *PDX1*, *HNF6*, *SOX9*, and *PTF1A*), as well as a population of early endocrine progenitor cells expressing endocrine TFs and genes (e.g., *NEUROG3*, *NEUROD1, FEV*, and *CHGA*) but lower levels of *PDX1* (Fig. 6c and Supplementary Fig. 7a). In control PP2 cultures, *NKX6.1* expression was restricted to multipotent pancreatic progenitors with high *PDX1* expression. By contrast, *NKX6.1* was broadly expressed in motif optimized cultures, including in cells expressing lower levels of *PDX1* (Fig. 6c, d and Supplementary Fig. 7b, c). Consistent with the lack of enhancer activation in motif optimized GT stage cells (Fig. 6b), there was no premature expression of *NKX6.1* at the GT stage (Supplementary Fig. 7d). These findings indicate that optimizing FOXA motif strength renders *NKX6.1* expression independent of high levels of PDX1. Corroborating this conclusion, we found NKX6.1 protein restricted to progenitors with high levels of PDX1 in control cultures, whereas motif optimized cultures contained a population of NKX6.1⁺/PDX1^low cells (Fig. 6e and Supplementary Fig. 7e). In sum, these findings show that increasing FOXA motif strength is sufficient to allow for FOXA recruitment independent of cooperative interactions with pancreatic lineage-determining TFs and that converting an unprimed into a primed enhancer lowers the target gene expression threshold (Fig. 6f).

Given that alpha cells are derived from NKX6.1⁻ endocrine progenitors, whereas beta cells arise from NKX6.1⁺ endocrine progenitors[30], we examined effects of broader *NKX6.1* expression among progenitors on cell fate allocation. To this end, we differentiated motif optimized and control cells to the early endocrine cell stage, when pre-alpha and pre-beta cells can be distinguished[29] (Supplementary Fig. 7f). We observed a two-fold increase in NKX6.1⁺/insulin⁺ cells accompanied by a decrease in glucagon expression (Supplementary Fig. 7g), suggesting a pre-alpha to a pre-beta cell fate shift. These results suggest that barriers to enhancer activation and target gene expression imposed by DNA sequence at class II enhancers are biologically relevant for cell lineage allocation during development.

**Distinct temporal patterns of FOXA1/2 occupancy distinguish hepatic and alveolar enhancers.** To determine whether the identified mechanisms of enhancer activation during organ development are universal across endodermal lineages, we also analyzed liver and lung enhancers, which like pancreatic enhancers undergo chromatin priming in gut endoderm[1]. Like pancreas

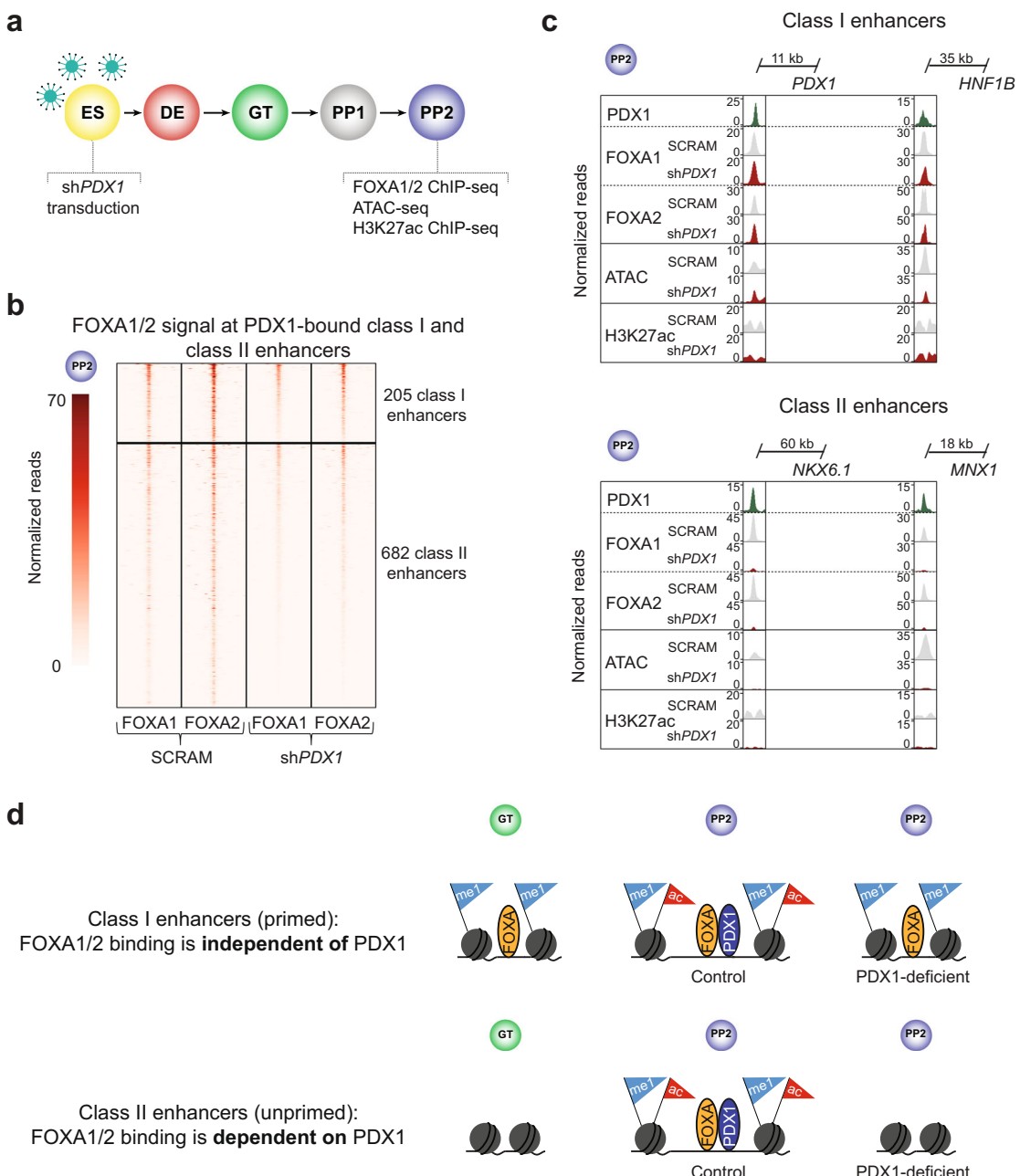

**Fig. 5 FOXA1/2 binding at class II enhancers is dependent on PDX1. a** Schematic of experimental design for *PDX1* knock-down in hESCs and subsequent differentiation into PP2 stage pancreatic progenitors. **b** Heatmap showing density of FOXA1 and FOXA2 ChIP-seq reads at PDX1-bound class I and class II pancreatic enhancers in hESCs transduced with scrambled control (SCRAM) or *PDX1* shRNA (sh*PDX1*) in PP2. Heatmap is centered on FOXA1 and FOXA2 peaks, respectively, and spans 5 kb. **c** Genome browser snapshots showing PDX1, FOXA1, and FOXA2 ChIP-seq, ATAC-seq, and H3K27ac ChIP-seq signal at class I enhancers near *PDX1* and *HNF1B* and class II enhancers near *NKX6.1* and *MNX1* in PP2. Approximate distance between enhancer and gene body is indicated. **d** Schematic illustrating distinct modes of FOXA TF recruitment at class I and class II pancreatic enhancers. FOXA1/2 recruitment depends on the lineage-determining TF PDX1 at class II enhancers. Both enhancer classes require PDX1 for activation. All ChIP-seq and ATAC-seq experiments, $n = 2$ replicates from independent differentiations.

development, both early liver and lung development depend on FOXA TFs[4–6]. Furthermore, previous studies have demonstrated FOXA binding to primed liver enhancers in gut endoderm prior to organ lineage induction[1,11]. To test whether class I and class II enhancers can be distinguished during liver and lung development, we induced the hepatic fate from hESC-GT stage intermediates (Fig. 7a), and generated distal lung alveolar epithelial type 2-like cells (iAT2s) grown at 95% purity as 3D alveolospheres (ALV) from iPSCs (Fig. 7b)[1,31]. For liver, we analyzed

H3K27ac signal and FOXA1/2 binding before liver induction at the GT stage and in hepatic progenitors (HP). For lung, we analyzed H3K27ac signal and FOXA1 binding before lung induction in anteriorized foregut (AFG) and at the ALV stage.

Analogous to the strategy used for identifying pancreatic enhancers (Supplementary Fig. 3d), we identified hepatic and alveolar enhancers based on gain in H3K27ac signal during the GT to HP and AFG to ALV transitions, respectively (≥2-fold change in H3K27ac, FDR ≤ 0.05; Supplementary Fig. 8a–d).

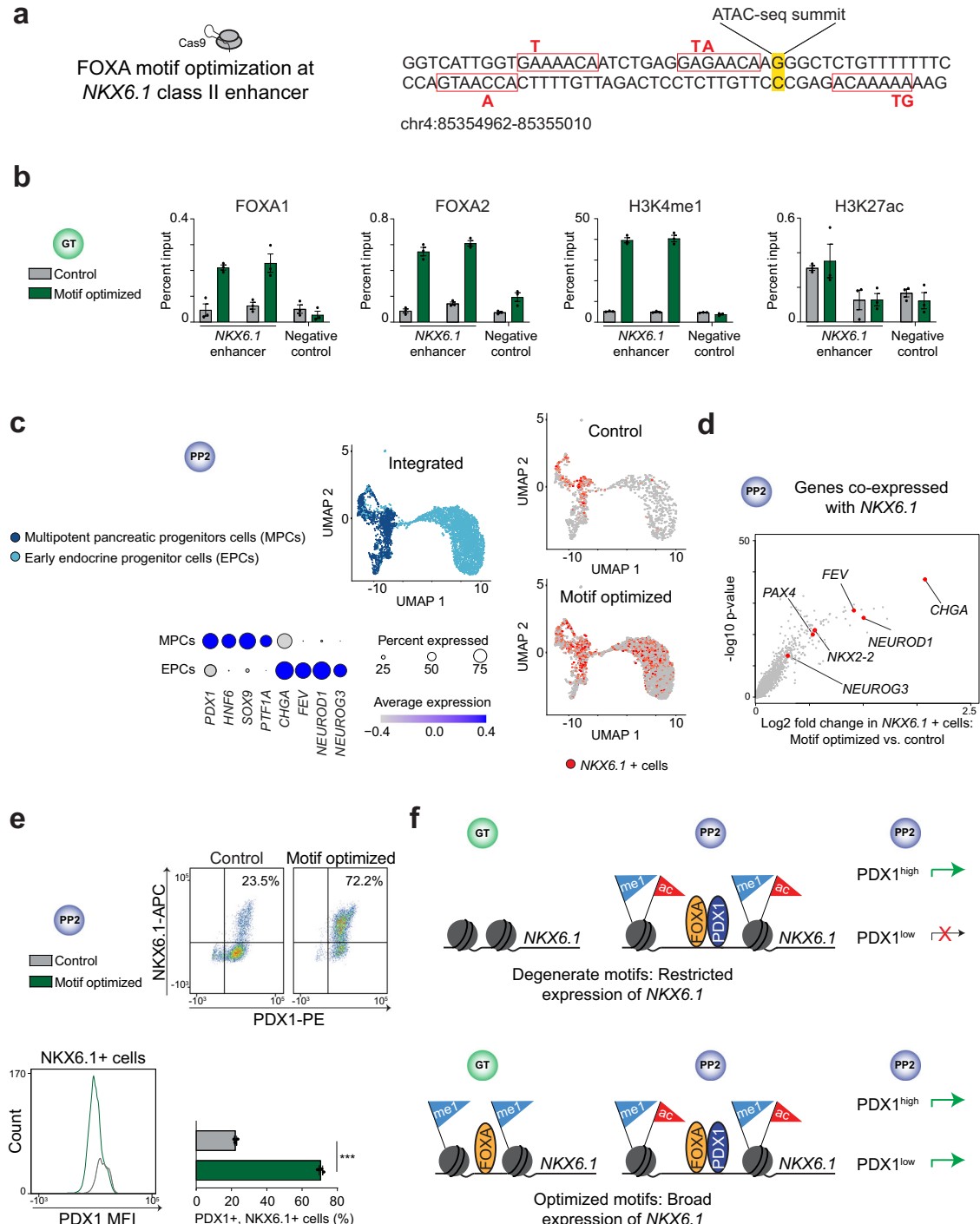

**Fig. 6 Optimization of FOXA-binding motifs at an *NKX6.1* enhancer redefines patterns of FOXA association and gene expression. a** Schematic illustrating base editing strategy at *NKX6.1* enhancer via CRISPR-Cas9. Degenerate FOXA-binding motifs and base edits are indicated in red. **b** ChIP-qPCR comparing FOXA1, FOXA2, H3K4me1, and H3K27ac ChIP-seq signal at the *NKX6.1* enhancer in control and motif optimized hESC lines at GT stage. Plots show two independent primer pairs for *NKX6.1* enhancer and one primer pair for a negative control region; $n = 3$ technical replicates. **c** UMAP representation of single-cell RNA-seq data from both control and motif optimized PP2 cells (integrated) and dot plot showing expression of marker genes in each population (bottom). *NKX6.1* expression across populations in control and motif optimized cell lines (right). **d** Volcano plot comparing genes co-expressed with *NKX6.1* in motif optimized compared to control PP2 cells. Wilcoxon rank sum test, 2-sided, corrected for multiple comparisons. **e** Representative flow cytometry analysis for PDX1 and NKX6.1, mean fluorescence intensity (MFI) of PDX1 signal in NKX6.1[+] cells, and quantification of PDX1[+] and NKX6.1[+] cells in control and motif optimized PP2 cells ($n = 3$ independent differentiations; $P < 1.0 \times 10^{-4}$; student's *t*-test, 2-sided). Bar graph shows mean ± S.E.M. **f** Schematic illustrating temporal patterns of FOXA recruitment and *NKX6.1* expression at the PP2 stage in cells with degenerate and optimized FOXA motifs at the *NKX6.1* enhancer.

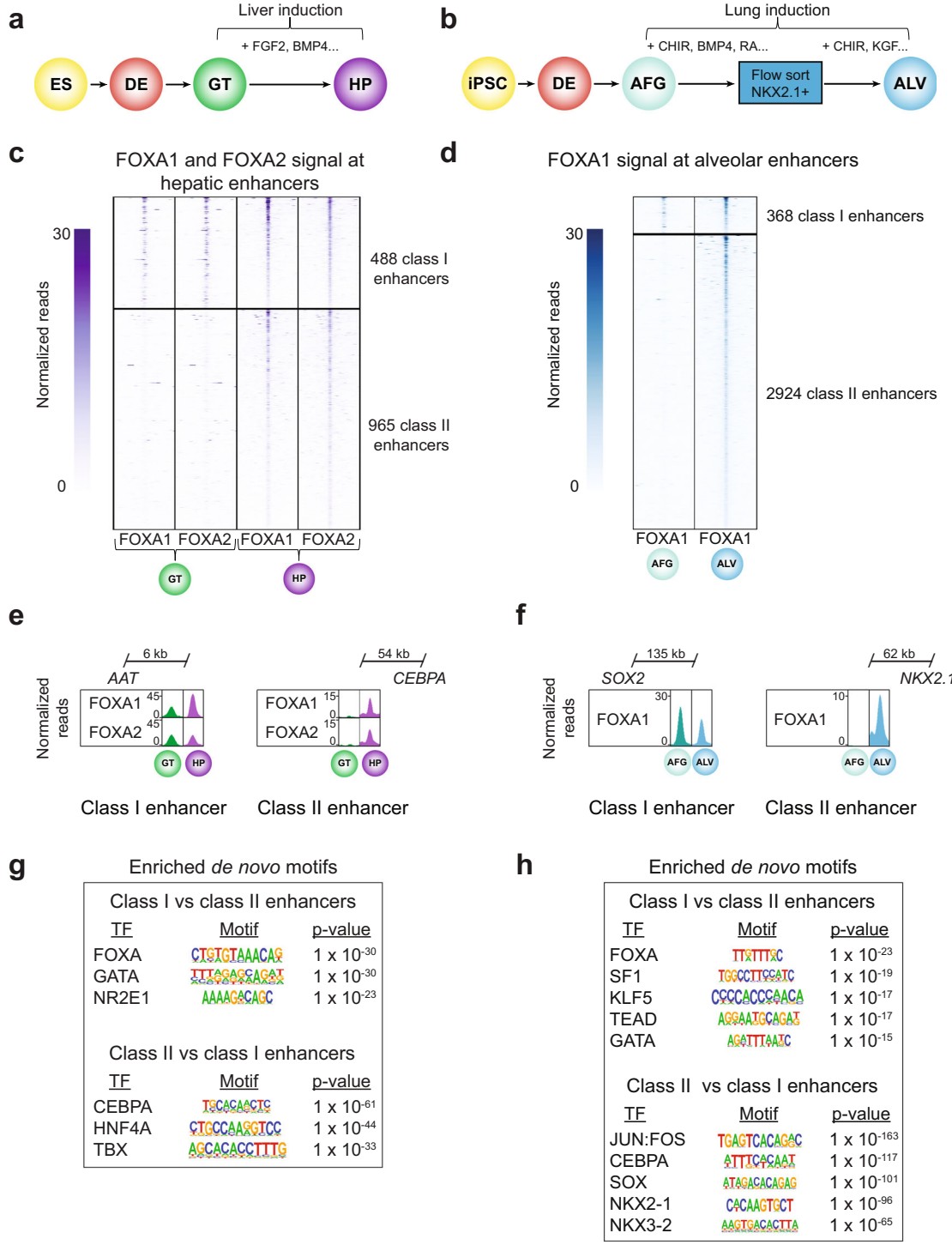

**Fig. 7 Class I and class II enhancers can be distinguished in liver and lung development. a** and **b** Schematic of stepwise differentiation of hESCs to hepatic progenitors (HP) (**a**) and induced human pluripotent stem cells (iPSC) into alveolosphere organoids (ALV) (**b**). AFG, anteriorized foregut. Select growth factors for hepatic (**a**) and alveolar (**b**) lineage induction are indicated. FGF2, fibroblast growth factor 2; BMP4, bone morphogenic protein 4; CHIR, CHIR99021; RA, retinoic acid. **c** Heatmap showing density of FOXA1 and FOXA2 ChIP-seq reads at hepatic enhancers in GT and HP. Heatmap is centered on FOXA1 and FOXA2 peaks, respectively, and spans 5 kb. Hepatic enhancers are classified based on temporal pattern of FOXA1 and FOXA2 occupancy. **d** Heatmap showing density of FOXA1 ChIP-seq reads at alveolar enhancers in AFG and ALV. Heatmap is centered on FOXA1 peaks and spans 5 kb. Alveolar enhancers are classified based on temporal pattern of FOXA1 occupancy. **e** Genome browser snapshots showing FOXA1 and FOXA2 ChIP-seq signal at a class I hepatic enhancer near *AAT* and a class II hepatic enhancer near *CEBPA* in GT and HP. **f** Genome browser snapshots showing FOXA1 ChIP-seq signal at a class I alveolar enhancer near *SOX2* and a class II alveolar enhancer near *NKX2.1* in AFG and ALV. **g** and **h** Enriched de novo transcription factor (TF)-binding motifs at class I against a background of class II enhancers and vice versa for hepatic (**g**) and alveolar enhancers (**h**). Fisher's exact test, 1-sided, corrected for multiple comparisons. All ChIP-seq experiments, $n = 2$ replicates from independent differentiations.

Subsequently, we quantified FOXA1/2 binding at the identified enhancers. As in pancreas, we observed two distinct patterns of FOXA1/2 occupancy (Fig. 7c, d and Supplementary Data 19–24) despite similar dynamics in H3K27ac signal (Supplementary Fig. 8e, f): a subset of class I enhancers exhibited FOXA1/2 occupancy prior to lineage induction (488 class I hepatic enhancers and 368 class I alveolar enhancers), whereas class II enhancers constituted the majority and exhibited de novo FOXA1/2 binding with lineage induction (965 class II hepatic enhancers and 2924 class II alveolar enhancers). These patterns were exemplified by enhancers near hepatic genes Alpha1-Antitrypsin (*AAT*) and *CEBPA* (Fig. 7e), as well as lung developmental TF genes *SOX2* and *NKX2.1* (Fig. 7f).

De novo motif analysis at class I against a background of class II hepatic enhancers revealed enrichment for FOXA motifs, GATA motifs, and the motif for the signal-dependent nuclear receptor NR2E1[32]. Class II enhancers showed comparative enrichment for motifs of the hepatic lineage-determining TFs CEBPA, HNF4A, and TBX[33,34] (Fig. 7g and Supplementary Data 25, 26), which increased in expression upon liver induction from hESC-GT intermediates (Supplementary Fig. 9a). FOXA2, HNF4A, and CEBP have been shown to co-bind liver-specific enhancers after liver induction[7], supporting a potential role for cooperative recruitment of FOXA TFs by these factors. Analogous to the motif enrichment patterns observed in pancreas and liver, alveolar class I enhancers were comparatively enriched for FOXA motifs, GATA motifs, and motifs for signal-dependent TFs NR5A1 (SF1) and TEAD with roles in lung development[35,36], whereas alveolar class II enhancers showed comparative motif enrichment for SOX family TFs and the lung master TF NKX2.1[37] (Fig. 7h and Supplementary Data 27 and 28). Thus, as in pancreas, a subset of hepatic and alveolar enhancers with canonical FOXA motifs and enrichment for motifs of signal-dependent TFs are FOXA1/2-bound prior to lineage induction, while de novo FOXA1/2 recruitment occurs at most hepatic and alveolar enhancers upon lineage induction.

To gain further insight into the architecture of hepatic and alveolar enhancers, we examined abundance, strength, and positioning of FOXA motifs. Using the same six FOXA PWMs as for pancreatic enhancers (Supplementary Fig. 5a), we observed significant enrichment for occurrence of FOXA motifs at both class I hepatic and class I alveolar enhancers (Supplementary Fig. 9b, c). We also found significantly higher log-odds scores for three FOXA PWMs (MA0047.2, MA0148.1, and MA0148.3; $P = 1.40 \times 10^{-3}$, $2.00 \times 10^{-3}$, and $1.60 \times 10^{-2}$, respectively; Wilcoxon rank sum test) at class I compared to class II hepatic enhancers, and two FOXA PWMs (MA0047.3 and MA0148.1; $P = 3.1 \times 10^{-2}$ and $4.1 \times 10^{-2}$, respectively; Wilcoxon rank sum test) at class I compared to class II alveolar enhancers. Furthermore, FOXA motif occurrence at ATAC-seq summits (444 and 701 ATAC-seq summits in class I and class II enhancers, respectively, at HP stage; Supplementary Fig. 9d) and occurrence of FOXA footprints (Supplementary Fig. 9e) were enriched at class I compared to class II hepatic enhancers. Thus, like pancreatic class I enhancers, hepatic and alveolar class I enhancers exhibit sequence features that have been associated with FOXA1/2 pioneering in other contexts[17,18]. Moreover, analogous to pancreatic enhancers, we observed preferential binding of GATA4 and GATA6 to class I compared to class II hepatic enhancers at the GT stage (Supplementary Fig. 9f), but no binding preference of the hepatic lineage-determining TF HNF4A at class II compared to class I hepatic enhancers despite HNF4A motif enrichment at HNF4A-bound class II enhancers (Supplementary Fig. 9f, g and Supplementary Data 29, 30). These results show that similar characteristics of sequence architecture distinguish pancreatic, hepatic, and alveolar class I and class II enhancers.

**Lineage-specific recruitment of FOXA1/2 to unprimed enhancers.** Our results suggest a model whereby the full enhancer complement for each endodermal organ lineage is established through (i) FOXA1/2-mediated priming of a small subset of enhancers for each lineage in endodermal precursors prior to lineage induction, and (ii) activation of a larger subset of unprimed enhancers by organ lineage-determining TFs that cooperatively recruit FOXA1/2 upon lineage induction. To determine the relationship between class I and class II enhancers across different endodermal lineages, we performed differential motif enrichment analysis, comparing class I or class II enhancers of each lineage against a background of class I or class II enhancers, respectively, of the alternate lineages. As expected, motifs for lineage-determining TFs for each lineage were enriched at both classes of enhancers (Supplementary Data 31–36). However, motif enrichment was stronger at class II than at class I enhancers (Fig. 8a), lending further support to the model that cooperativity with lineage-determining TFs facilitates lineage-specific FOXA1/2 association with class II enhancers of each organ. Consistent with the binding of FOXA1/2 to class I enhancers in shared developmental precursors prior to lineage induction, we found that class I enhancers of one organ lineage were more frequently bound by FOXA1/2 in alternate lineages than class II enhancers (Fig. 8b, c). Altogether, these findings support establishment of organ-specific gene expression programs through two distinct mechanisms of FOXA1/2-mediated enhancer activation (Fig. 8d).

## Discussion

FOXA TFs are generally thought to control developmental transitions by mediating chromatin priming owing to FOXA's pioneer TF activity[1,11,38]. We have previously reported that chromatin priming and FOXA1/2 recruitment precede organ lineage induction at pancreas, liver, and lung enhancers[1]. Here, we show that chromatin priming and early FOXA1/2 recruitment are limited to a small subset of organ lineage enhancers, whereas the majority transitions from unprimed to active and engages FOXA1/2 upon lineage induction. We demonstrate that DNA sequence logic is the primary determinant of whether an enhancer is primed and recruits FOXA1/2 independent of lineage-specific TFs or whether it is unprimed and requires lineage-specific TFs for FOXA1/2 binding. The results presented here provide a molecular framework for understanding gene regulatory principles that underlie lineage induction and cell type diversification during organogenesis. Our findings support a model whereby FOXA-mediated priming of a subset of organ-specific enhancers enables the initiation of organ-specific gene expression programs by lineage-inductive cues, whereas secondary recruitment of FOXA by lineage-specific TFs to most organ-specific enhancers helps establish cell type-specific gene expression by safeguarding against broad target gene expression within the organ progenitor domain.

We observed stronger and more abundant FOXA motifs at primed compared to unprimed enhancers and found that FOXA1/2 recruitment to a proportion of unprimed enhancers depends on the pancreatic TF PDX1. Furthermore, we show that strengthening FOXA motifs at an unprimed enhancer obviates dependency of FOXA1/2 binding on PDX1, resulting in FOXA recruitment and enhancer priming prior to lineage induction. Our findings are consistent with prior observations in tumor cell line models, which have suggested that the ability of FOXA TFs to stably bind and remodel chromatin is DNA sequence-dependent[17,18,20]. Our results extend these observations in immortalized cell lines to demonstrate relevance of distinct mechanisms of FOXA recruitment for developmental gene regulation.

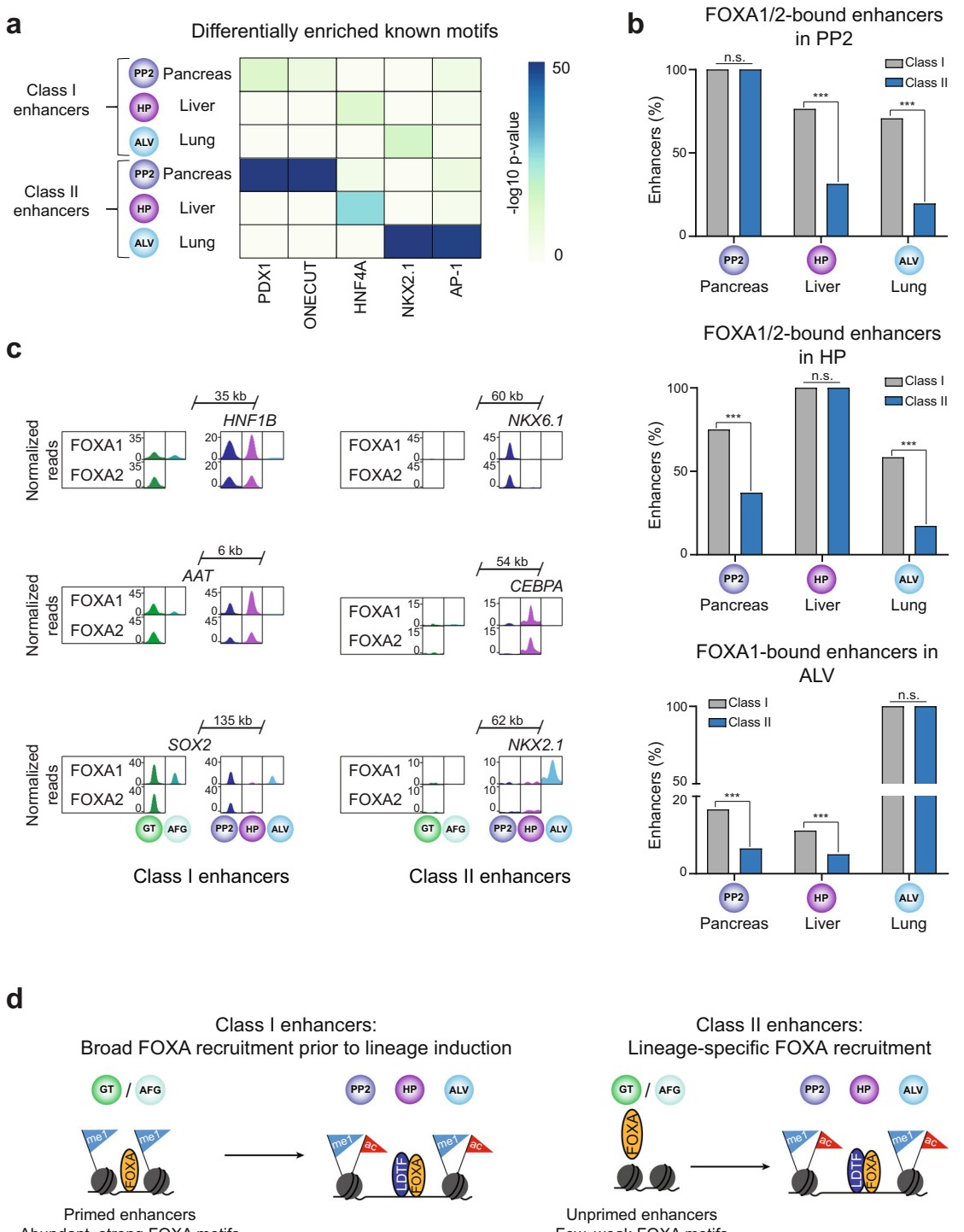

**Fig. 8 Recruitment of FOXA1/2 to class II enhancers is lineage-specific. a** Heatmap showing enrichment of known binding motifs for lineage-determining transcription factors at pancreatic, hepatic, and alveolar class I and class II enhancers. Class I and class II enhancers of each lineage were compared against a background of class I and class II enhancers, respectively, of all other lineages. Fisher's exact test, 1-sided, corrected for multiple comparisons. **b** Percentage of pancreatic, hepatic, and alveolar class I and class II enhancers overlapping FOXA1/2 ChIP-seq peaks (within 100 bp from peak) in PP2 (pancreas), HP (liver) and ALV (lung). For ALV only FOXA1 peaks were considered. $P = 1$, $<2.2 \times 10^{-16}$, $<2.2 \times 10^{-16}$, $<2.2 \times 10^{-16}$, $1$, $<2.2 \times 10^{-16}$, $5.86 \times 10^{-11}$, $3.08 \times 10^{-5}$, and 1 for comparisons of FOXA occupancy at class I and class II pancreatic, hepatic, and alveolar enhancers at PP2, HP, and ALV stage cells, respectively; Fisher's exact test, 2-sided. **c** Genome browser snapshots showing FOXA1/2 ChIP-seq signal across endodermal lineages at example pancreatic, hepatic, and alveolar class I and class II enhancers. Approximate distance between enhancer and gene body is indicated. **d** Schematic showing differential recruitment of FOXA TFs to endodermal organ class I and class II enhancers during endoderm development. LDTF, lineage-determining transcription factor. All ChIP-seq experiments, $n = 2$ replicates from independent differentiations.

Our observation that FOXA1/2 bind primed enhancers without cooperative recruitment by pancreatic TFs raises the question of how FOXA TFs engage their target sites at primed enhancers. We find that a subset of primed enhancers is bound by both FOXA and GATA TFs prior to lineage induction. Given previously demonstrated cooperativity between FOXA and GATA TFs[17], it is possible that GATA TFs help recruit FOXA to a subset of primed enhancers. However, we show that strengthening FOXA motifs is sufficient to enable FOXA1/2 binding to an enhancer not bound by GATA TFs. Therefore, our data support the conclusion that strong FOXA motifs are sufficient to facilitate FOXA TF engagement and chromatin priming during development, consistent with observations that FOXA1/2 can engage target sites on nucleosomal DNA in vitro[7−9].

Our findings provide insight into the gene regulatory mechanisms that underlie endodermal organ lineage induction and cell fate specification. We observed enrichment of binding motifs for signal-dependent TFs and binding of the retinoic acid receptor subunit RXR at primed pancreatic enhancers. These findings suggest that organ lineage-inductive cues are read by primed enhancers to initiate expression of lineage-determining TFs. In support of this, primed enhancers are found near PDX1, HNF1B, and MEIS1, which are among the first TFs expressed upon pancreas induction. By contrast, unprimed enhancers are enriched for binding motifs of organ-specific TFs, which recruit FOXA1/2 secondarily. Given that FOXA TFs are broadly expressed across endodermal organ lineages, indirect FOXA recruitment by organ-specific TFs provides a safeguard against lineage-aberrant enhancer activation and gene expression. This agrees with studies in Drosophila and Ciona, which suggest that suboptimization of TF-binding motifs could be a general principle by which to confer cell specificity to enhancers[26,39].

Replacing low-affinity FOXA-binding sites at an unprimed enhancer for NKX6.1 with higher affinity sites broadened the domain of NKX6.1 expression among pancreatic progenitors. As we show, NKX6.1 was not prematurely expressed, demonstrating that motif optimization does not eliminate the dependency of target gene expression on lineage-specific cues. This suggests that early FOXA recruitment through high affinity-binding sites lowers the threshold for target gene expression, which could reflect an increased sensitivity of the enhancer to activation by lineage-specific TFs. Thus, higher thresholds to target gene expression conferred by unprimed enhancers will restrict target gene expression to specific cell populations, as enhancer activation will only occur when a specific complement of lineage-specific TFs is present in sufficient concentrations. We propose that gene regulation by unprimed enhancers provides a mechanism for specifying different cell types early in organ development. Small differences in TF expression among early organ progenitors would be sufficient to activate different repertoires of unprimed enhancers, thereby creating divergent gene expression patterns and cell populations. Consistent with this concept, it has been shown that PDX1high and PDX1low cells in the early pancreatic epithelium acquire different cell identities[40].

We demonstrate that conversion of a single enhancer near NKX6.1 from an unprimed to a primed state is sufficient to alter cell fate due to broadened expression of NKX6.1 within the progenitor cell domain. These findings show that in a developmental context, differences in FOXA-binding affinity at enhancers can affect cell fate allocation. It is therefore possible that polymorphisms at FOXA-binding sites determine interindividual differences in endodermal organ cell type composition. Consistent with this possibility, islet cell type composition is known to vary greatly in humans[41] and the NKX6.1 enhancer contains twelve known polymorphisms predicted to alter the strength and spacing of FOXA motifs. While the importance of polymorphisms for organ cell type composition remains to be demonstrated, our findings support the concept that FOXA TF motif strength at developmental enhancers provides a tunable threshold for target gene expression.

## Methods

### Cell lines and animal models

*Human cell culture experiments.* hESC research was approved by the University of California, San Diego (UCSD), Institutional Review Board and Embryonic Stem Cell Research Oversight Committee (protocol 090165ZX). Human iPSC research was approved by the Boston University Institutional Review Board (protocol H-33122).

*Maintenance of HEK293T cells.* HEK293T cells (female) were cultured in a humidified incubator at 37 °C with 5% $CO_2$ using Dulbecco's Modified Eagle Medium (Cat# 45000-312; 4.5 g/L glucose, [+] L-glutamine, [-] sodium pyruvate) supplemented with 10% fetal bovine serum (FBS) and 1% Penicillin-Streptomycin (Thermo Fisher Scientific, Cat# 15140122).

*Maintenance and differentiation of CyT49 hESCs.* CyT49 hESCs (male) were maintained and differentiated as described[1,42,43]. Propagation of CyT49 hESCs was carried out by passing cells every 3 to 4 days using Accutase™ (eBioscience) for enzymatic cell dissociation, and with 10% (v/v) human AB serum (Valley Biomedical) included in the hESC media the day of passage. hESCs were seeded into tissue culture flasks at a density of 50,000 cells/cm². hESC media was comprised of DMEM/F12 (VWR) supplemented with 10% (vol/vol) KnockOut™ Serum Replacement XenoFree (Life Technologies), 0.1 mM MEM non-essential amino acids (Life Technologies), 1X GlutaMAX™ I (Life Technologies), 1% (vol/vol) penicillin/ streptomycin (Life Technologies), 0.1 mM 2-mercaptoethanol (Life Technologies), 10 ng/mL Activin A (R&D Systems), and 10 ng/mL Heregulin-β1 (PeproTech).

Pancreatic differentiation was performed as previously described[1,42,43]. Briefly, a suspension-based culture format was used to differentiate cells in aggregate form. Undifferentiated aggregates of hESCs were formed by re-suspending dissociated cells in hESC maintenance medium at a concentration of $1 \times 10^6$ cells/mL and plating 5.5 mL per well of the cell suspension in 6-well ultra-low attachment plates (Costar). The cells were cultured overnight on an orbital rotator (Innova2000, New Brunswick Scientific) at 95 rpm (0.2 x g). After 24 h the undifferentiated aggregates were washed once with RPMI medium and supplied with 5.5 mL of day 0 differentiation medium. Thereafter, cells were supplied with the fresh medium for the appropriate day of differentiation (see below). Cells were continually rotated at 95 rpm (0.2 x g), or 105 rpm (0.2 x g) on days 4 through 8, and no media change was performed on day 10. Both RPMI (Mediatech) and DMEM High Glucose (HyClone) medium were supplemented with 1X GlutaMAX™ and 1% penicillin/ streptomycin. Human activin A, mouse Wnt3a, human KGF, human noggin, and human EGF were purchased from R&D systems. Other added components included FBS (HyClone), B-27® supplement (Life Technologies), Insulin-Transferrin-Selenium (ITS; Life Technologies), TGFβ R1 kinase inhibitor IV (EMD Bioscience), KAAD-Cyclopamine (KC; Toronto Research Chemicals), and the retinoic receptor agonist TTNPB (RA; Sigma Aldrich). Day-specific differentiation media formulations were as follows:

Days 0 and 1: RPMI + 0.2% (v/v) FBS, 100 ng/mL Activin, 50 ng/mL mouse Wnt3a, 1:5000 ITS. Days 1 and 2: RPMI + 0.2% (v/v) FBS, 100 ng/mL Activin, 1:5000 ITS

Days 2 and 3: RPMI + 0.2% (v/v) FBS, 2.5 mM TGFβ R1 kinase inhibitor IV, 25 ng/mL KGF, 1:1000 ITS

Days 3−5: RPMI + 0.2% (v/v) FBS, 25 ng/mL KGF, 1:1000 ITS

Days 5−8: DMEM + 0.5X B-27® Supplement, 3 nM TTNPB, 0.25 mM KAAD-Cyclopamine, 50 ng/mL Noggin

Days 8−10: DMEM/B-27, 50 ng/mL KGF, 50 ng/mL EGF

Cells at D0 correspond to the embryonic stem cell (ES) stage, cells at D2 correspond to the definitive endoderm (DE) stage, cells at D5 correspond to the gut tube (GT) stage, cells at D7 correspond to the early pancreatic progenitor (PP1) stage, and cells at D10 correspond to the late pancreatic progenitor (PP2) stage.

Hepatic differentiation was performed as previously described[1]. Briefly, cells were treated identically as in pancreatic differentiation until the GT stage at D5. At this point cells were treated with 50 ng/mL BMP4 (Millipore) and 10 ng/mL FGF2 (Millipore) in RPMI media (Mediatech) supplemented with 0.2% (vol/vol) FBS (HyClone) for 3 days with daily media changes. Cells at D8 correspond to the hepatic progenitor (HP) cell stage. A full list of reagents and catalog numbers is provided in Supplementary Table 1.

*Maintenance and differentiation of H1 hESCs.* H1 hESCs (male) were maintained and differentiated as described with some modifications[44,45]. In brief, hESCs were cultured in mTeSR1 media (Stem Cell Technologies) supplemented with 1% Penicillin-Streptomycin (Thermo Fisher Scientific, Cat# 15140122) and propagated by passaging cells onto 6-well plates coated with Matrigel (Corning) every 3 to 4 days using Accutase (eBioscience) for enzymatic cell dissociation.

For differentiation, cells were dissociated using Accutase for 10 min, then reaggregated in mTESR supplemented with Y-27632 (Stem Cell Technologies) by plating the cells at a concentration of ~5.5 × 10⁶ cells/well in a low attachment 6-well plate on an orbital shaker (100 rpm, 0.2 x g) in a 37 °C incubator. The following day, undifferentiated cells were washed in base media (see below) and then differentiated using a multi-step protocol with stage-specific media and daily media changes.

All stage-specific base media were comprised of MCDB 131 medium (Thermo Fisher Scientific) supplemented with NaHCO3, GlutaMAX, D-Glucose, and BSA using the following concentrations:

Stage 1/2 base medium: MCDB 131 medium, 1.5 g/L NaHCO3, 1X GlutaMAX, 10 mM D-Glucose, 0.5% BSA

Stage 3/4 base medium: MCDB 131 medium, 2.5 g/L NaHCO3, 1X GlutaMAX, 10 mM D-glucose, 2% BSA

Stage 5 medium: MCDB 131 medium, 1.5 g/L NaHCO3, 1X GlutaMAX, 20 mM D-glucose, 2% BSA

Media compositions for each stage were as follows:

Stage 1 (days 0−2): base medium, 100 ng/mL Activin A, 25 ng/mL Wnt3a (day 0). Day 1–2: base medium, 100 ng/mL Activin A

Stage 2 (days 3−5): base medium, 0.25 mM L-Ascorbic Acid (Vitamin C), 50 ng/mL FGF7

Stage 3 (days 6−7): base medium, 0.25 mM L-Ascorbic Acid, 50 ng/mL FGF7, 0.25 μM SANT-1, 1 μM Retinoic Acid, 100 nM LDN193189, 1:200 ITS-X, 200 nM TPB

Stage 4 (days 8−10): base medium, 0.25 mM L-Ascorbic Acid, 2 ng/mL FGF7, 0.25 μM SANT-1, 0.1 μM Retinoic Acid, 200 nM LDN193189, 1:200 ITS-X, 100 nM TPB

Stage 5 (days 11−13): base medium, 0.25 μM SANT-1, 0.05 μM RA, 100 nM LDN-193189, 1 μM T3, 10 μM ALK5i II, 10 μM ZnSO4, 10 μg/mL heparin, 1:200 ITS-X

Cells at D0, D3, D6, D8, D11, and D14 correspond to the ES DE, GT, PP1, PP2, and EN stages, respectively. At D8 of differentiation, speed of the orbital shaker was increased to 110 rpm (0.3 x g). A full list of reagents and catalog numbers is provided in Supplementary Table 2.

*Maintenance and differentiation of iPSCs.* SPC2 iPSCs (male; clone SPC2-ST-B2[46]) were maintained in feeder-free culture conditions in 6-well tissue culture dishes (Corning) coated with growth factor-reduced Matrigel (Corning, Cat# 356231), in mTeSR1 medium (Stem Cell Technologies, Cat# 85850) and passaged using gentle cell dissociation reagent (GCDR; Stem Cell Technologies, Cat# 07174). Details of iPSC derivation, characterization, and differentiation into anterior foregut endoderm and alveolar epithelial type 2 cells (iAT2s; also known as iAEC2s) have been previously published[31,46,47] and are available for free download at http://www.bu.edu/dbin/ stemcells/protocols.php. Briefly, the SPC2-ST-B2 iPSC clone, engineered to carry a tdTomato reporter knocked into one allele of the endogenous *SFTPC* locus[46], underwent directed differentiation to generate iAT2s in 3D Matrigel cultures as follows. Cells were first differentiated into definitive endoderm using the STEMdiff Definitive Endoderm Kit (Stem Cell Technologies, Cat# 05110) for 72 h and subsequently dissociated with GCDR and passaged as small clumps into growth factor-reduced Matrigel-coated (Corning, Cat# 356231) 6-well culture plates (Corning) in "DS/SB" foregut endoderm anteriorization media, consisting of complete serum-free differentiation medium (cSFDM) base as previously described[31], supplemented with 10 μm SB431542 ("SB"; Tocris, Cat# 1614) and 2 μm Dorsomorphin ("DS"; Stemgent, Cat# 04-0024), to pattern cells towards anterior foregut endoderm (AFE; day 6 of differentiation). For the first 24 h after passaging, media was supplemented with 10 μM Y-27632 (Stem Cell Technologies, Cat# 72305). After anteriorization in DS/SB media for 72 h, beginning on day 6 of differentiation cells were cultured in "CBRa" lung progenitor-induction medium for 9 additional days. "CBRa" medium consists of cSFDM base supplemented with 3 μM CHIR99021 (Tocris, Cat# 4423), 10 ng/mL recombinant human BMP4 (rhBMP4; R&D Systems, Cat#314-BP), and 100 nM retinoic acid (RA; Sigma, Cat# R2625), as described[31]. On differentiation day 15, NKX2-1⁺ lung progenitors were isolated based on CD47hi/CD26neg gating[48] using a high-speed cell sorter (MoFlo Legacy or MoFlo Astrios EQ). Purified day 15 lung progenitors were resuspended in undiluted growth factor-reduced 3D Matrigel (Corning, Cat# 356231) at a concentration of 400 cells/μL and distal/alveolar differentiation was performed in "CK + DCI" medium, consisting of cSFDM base supplemented with 3 μm CHIR99021 (Tocris, Cat# 4423), 10 ng/mL rhKGF (R&D Systems, Cat# 251-KG), and 50 nM dexamethasone (Sigma, Cat# D4902), 0.1 mM 8-Bromoadenosine 30, 50-cyclic monophosphate sodium salt (Sigma, Cat# B7880) and 0.1 mM 3-Isobutyl-1-methylxanthine (IBMX; Sigma, Cat# I5879) (DCI) with a brief period of CHIR99021 withdrawal between days 34-39 to achieve iAT2 maturation. To establish pure cultures of iAT2s, cells were sorted by flow cytometry on day 45 to purify SFTPC^tdTomato+ cells. iAT2s were maintained as self-renewing monolayered epithelial spheres ("alveolospheres") through serial passaging every 10-14 days and replating in undiluted growth factor-reduced 3D Matrigel (Corning, Cat# 356231) droplets at a density of 400 cells/μl in CK + DCI medium, as described[47]. iAT2 culture quality and purity was monitored at each passage by flow cytometry, with 95.2 ± 4.2% (mean ± S.D.) of cells expressing SFTPC^tdTomato over time, as we have previously detailed[31,46].

Cells at day 6 correspond to the AFG stage and day 261 iAT2s were used for the alveolar stage.

**Generation of FOXA1⁻/⁻, FOXA2⁻/⁻, and FOXA1/2⁻/⁻ H1 hESC lines.** To generate homozygous *FOXA1, FOXA2*, and *FOXA1/2* deletion hESC lines, sgRNAs targeting coding exons within each gene were cloned into Px333-GFP, a modified version of Px333[49], which was a gift from Andrea Ventura (Addgene, #64073). The plasmid was transfected into H1 hESCs with XtremeGene 9 (Roche, Cat# 6365787001), and 24 h later 8000 GFP⁺ cells were sorted into a well of six-well plate. Individual colonies that emerged within 5–7 days were subsequently transferred manually into 48-well plates for expansion, genomic DNA extraction, PCR genotyping, and Sanger sequencing. For control clones, the Px333-GFP plasmid was transfected into H1 hESCs, and cells were subjected to the same workflow as H1 hESCs transfected with sgRNAs.

sgRNA oligo used to generate *FOXA1⁻/⁻* hESCs: CGCCATGAACAGCATGACTG
sgRNA oligo used to generate *FOXA2⁻/⁻* hESCs: CATGAACATGTCGTCGTACG
sgRNA oligos used to generate *FOXA1/2⁻/⁻* frameshift hESCs:
*FOXA1*: CGCCATGAACAGCATGACTG
*FOXA2*: CATGAACATGTCGTCGTACG
sgRNA oligos used to generate *FOXA1/2⁻/⁻* exon deletion hESCs:
*FOXA1* upstream: GCGACTGGAACAGCTACTAC
*FOXA1* downstream: GCACTGCAATACTCGCCTTA
*FOXA2* upstream: TCCGACTGGAGCAGCTACTA
*FOXA2* downstream: CGGCTACGGTTCCCCCATGC

**Generation of NKX6.1 enhancer motif optimized H1 hESC line.** To generate base substitutions in the *NKX6.1* enhancer, a sgRNA targeting the enhancer was cloned into the Px458 plasmid[50], which was a gift from Feng Zhang (Addgene, #48138). The plasmid and an asymmetric single-stranded oligodeoxynucleotide donor template (ssODN) were transfected into H1 hESCs with XtremeGene 9 (Roche, Cat# 6365787001), and cells were treated with 1 μM SCR7 DNA ligase IV inhibitor (Stem Cell Technologies, Cat# 74102) to promote homology-directed repair. Twenty-four hours later 8000 GFP⁺ cells were sorted into a well of six-well plate. Individual colonies that emerged within 5–7 days were subsequently transferred manually into 48-well plates for expansion, genomic DNA extraction, PCR genotyping, and Sanger sequencing.

sgRNA oligo used to target *NKX6.1* enhancer: GAAGCTCTCTACCTAGTGTG
ssODN sequence:

TGCCTATGATTTATGTATTTGTTTAGTCAATAGTCTAATGTAAATGATGT
AATTAATTATAGATGGTGGTGTCAGGTCATTTGTGTAAACAATCTGAGG
TAAACAAGGGCTCTGTTTACTTCATGACAGATGCAGGGGGGTGGGGGGC
TGAGTTGAGGGAATTCCAGGGGAACTTTTTCACGTGTGAATGGCGGCTG
GGA

**Transduction of CyT49 hESCs with SCRAM and shPDX1.** To generate shRNA expression vectors, shRNA guide sequences were placed under the control of the human U6 pol III promoter in the pLL3.7 backbone[51], which was a gift from Luk Parijs (Addgene, plasmid #11795). Short hairpin sequences are provided in Supplementary Table 3.

High-titer lentiviral supernatants were generated by co-transfection of the shRNA expression vector and the lentiviral packaging construct into HEK293T cells as described[42]. Briefly, shRNA expression vectors were co-transfected with the pCMV-R8.74 and pMD2.G expression plasmids (Addgene #22036 and #12259, respectively, gifts from Didier Trono) into HEK293T cells using a 1 mg/mL PEI solution (Polysciences, Cat# 23966-1). Lentiviral supernatants were collected at 48 h and 72 h after transfection. Lentiviruses were concentrated by ultracentrifugation for 120 min at 68,567 x g using a Beckman SW28 ultracentrifuge rotor at 4 °C.

CyT49 hESCs were plated onto a six-well plate at a density of 1 million cells per well. The following morning, concentrated lentivirus was added at 5 μL/mL media, as well as 8 μg/mL polybrene (Fisher Scientific, Cat# TR1003G). After 30 min of incubation, the 6-well plate was spun in a centrifuge (Sorvall Legend RT) for 1 h at 30 °C at 950 x g. 6 h later, viral media was replaced with fresh base culture media. After 72 h, cells were sorted for GFP expression and re-cultured.

**Immunofluorescence analysis.** Cell aggregates derived from hESCs were allowed to settle in microcentrifuge tubes and washed twice with PBS before fixation with 4% paraformaldehyde (PFA) for 30 min at room temperature. Fixed samples were washed twice with PBS and incubated overnight at 4 °C in 30% (w/v) sucrose in PBS. Samples were then loaded into disposable embedding molds (VWR), covered in Tissue-Tek® O.C.T. Sakura® Finetek compound (VWR) and flash frozen on dry ice to prepare frozen blocks. The blocks were sectioned at 10 μm and sections were placed on Superfrost Plus® (Thermo Fisher) microscope slides and washed with PBS for 10 min. Slide-mounted cell sections were permeabilized and blocked with blocking buffer, consisting of 0.15% (v/v) Triton X-100 (Sigma, Cat# T8787) and 1% (v/v) normal donkey serum (Jackson Immuno Research Laboratories, Cat# 017-000-121) in PBS, for 1 h at room temperature. Slides were then incubated overnight at 4 °C with primary antibody solutions. The following day slides were washed five times with PBS and incubated for 1 h at room temperature with secondary antibody solutions. Cells were washed five times with PBS before coverslips were applied.

All antibodies were diluted in blocking buffer at the ratios indicated below. Primary antibodies used were mouse anti-FOXA1 (1:100 or 1:1000 dilution, Abcam ab55178); goat anti-FOXA2 (1:300 dilution, R&D systems AF2400); goat anti-SOX17 (1:300 dilution, R&D systems AF1924); goat anti-HNF4A (1:1000 dilution, Santa Cruz Biotechnology SC-6556); rabbit anti-PDX1 (1:500 dilution, Abcam ab47267); and mouse anti-NKX6.1 (1:300 dilution, Developmental Studies Hybridoma Bank F64A6B4). Secondary antibodies against mouse, rabbit, and goat were Alexa488- and Cy3-conjugated donkey antibodies (Jackson Immuno Research Laboratories, Cat# 715-165-150, 711-485-152, and 705-545-003, respectively), and were used at dilutions of 1:500 (anti-rabbit Alexa488) or 1:1000 (all other secondary antibodies). Cell nuclei were stained with Hoechst 33342 (1:3000, Invitrogen, Cat# H3570). Representative images were obtained with a Zeiss Axio-Observer-Z1 microscope equipped with a Zeiss ApoTome and AxioCam digital camera. Figures were prepared in Adobe Creative Suite 5.

**Flow cytometry analysis.** Cell aggregates derived from hESCs were allowed to settle in microcentrifuge tubes and washed with PBS. Cell aggregates were incubated with Accutase® at 37 °C until a single-cell suspension was obtained. Cells were washed with 1 mL ice-cold flow buffer comprised of 0.2% BSA in PBS and centrifuged at 200 x $g$ for 5 min. BD Cytofix/Cytoperm™ Plus Fixation/Permeabilization Solution Kit was used to fix and stain cells for flow cytometry according to the manufacturer's instructions. Briefly, cell pellets were resuspended in ice-cold BD Fixation/Permeabilization solution (300 μL per microcentrifuge tube). Cells were incubated for 20 min at 4 °C. Cells were washed twice with 1 mL ice-cold 1X BD Perm/Wash™ Buffer and centrifuged at 10 °C and 200 x $g$ for 5 min. Cells were resuspended in 50 μL ice-cold 1X BD Perm/Wash™ Buffer containing diluted antibodies, for each staining performed. Cells were incubated at 4 °C in the dark for 1–3 h. Cells were washed with 1.25 mL ice-cold 1X BD Wash Buffer and centrifuged at 200 x $g$ for 5 min. Cell pellets were resuspended in 300 μL ice-cold flow buffer and analyzed in a FACSCanto II (BD Biosciences). Antibodies used were PE-conjugated anti-SOX17 antibody (1:20 dilution, BD Biosciences AF1924); mouse anti-HNF1B antibody (1:100 dilution, Santa Cruz Biotechnology sc-130407); PE-conjugated anti-mouse IgG (1:50 dilution, BD Biosciences 555749); PE-conjugated anti-PDX1 (1:10 dilution, BD Biosciences 562161); AlexaFluor® 647-conjugated anti-NKX6.1 (1:5 dilution, BD Biosciences 563338); and PE-conjugated anti-Insulin (1:50 dilution, Cell Signaling 8508). Data were processed using FlowJo software v10.

**Chromatin immunoprecipitation sequencing (ChIP-seq).** ChIP-seq was performed using the ChIP-IT High-Sensitivity kit (Active Motif) according to the manufacturer's instructions. Briefly, for each cell stage and condition analyzed, 5–10 × 10$^6$ cells were harvested and fixed for 15 min in an 11.1% formaldehyde solution. Cells were lysed and homogenized using a Dounce homogenizer and the lysate was sonicated in a Bioruptor® Plus (Diagenode), on high for 3 × 5 min (30 s on, 30 s off). Between 10 and 30 μg of the resulting sheared chromatin was used for each immunoprecipitation. Equal quantities of sheared chromatin from each sample were used for immunoprecipitations carried out at the same time. Four micrograms of antibody were used for each ChIP-seq assay. Chromatin was incubated with primary antibodies overnight at 4 °C on a rotator followed by incubation with Protein G agarose beads for 3 h at 4 °C on a rotator. Antibodies used were rabbit anti-H3K27ac (Active Motif 39133); rabbit anti-H3K4me1 (Abcam ab8895); goat anti-FOXA1 (Abcam Ab5089); goat-anti-FOXA2 (Santa Cruz SC-6554); goat anti-GATA4 (Santa Cruz SC-1237); mouse anti-GATA6 (Santa Cruz SC-9055); and mouse anti-HNF4A (Novus PP-H1415). Reversal of crosslinks and DNA purification were performed according to the ChIP-IT High-Sensitivity instructions, with the modification of incubation at 65 °C for 2-3 h, rather than at 80 °C for 2 h. Sequencing libraries were constructed using KAPA DNA Library Preparation Kits for Illumina® (Kapa Biosystems) and library sequencing was performed on either a HiSeq 4000 System (Illumina®) or NovaSeq 6000 System (Illumina®) with single-end reads of either 50 or 75 base pairs (bp). Sequencing was performed by the UCSD Institute for Genomic Medicine (IGM) core research facility. For ChIP-seq experiments at the DE, AFG, and ALV stages in iAEC2 cells, two technical replicates from a single differentiation were generated. For all other ChIP-seq experiments, replicates from two independent hESC differentiations were generated.

**ChIP-qPCR.** For ChIP-qPCR, immunoprecipitation, reversal of crosslinks, and DNA purification were performed as for ChIP-seq. Antibodies used were rabbit anti-H3K27ac (Active Motif 39133); rabbit anti-H3K4me1 (Abcam ab8895); goat anti-FOXA1 (Abcam Ab5089); and goat anti-FOXA2 (R&D AF2400). After DNA purification, each sample and a 1% dilution of input DNA used for immunoprecipitation were amplified using 2 independent primers targeting either the histones flanking the *NKX6.1* enhancer (for measurements of H3K4me1 and H3K27ac) or the FOXA-binding site (for measurements of FOXA1 and FOXA2), as well as a negative control region. qPCR reactions were performed in technical triplicates using a CFX96™ Real-Time PCR Detection System and the iQ™ SYBR® Green Supermix (Bio-Rad, Cat# 1708880). A complete list of primer sequences is provided in Supplementary Table 4.

**ChIP-seq data analysis.** ChIP-seq reads were mapped to the human genome consensus build (hg19/GRCh37) and visualized using the UCSC Genome Browser[52]. Burrows-Wheeler Aligner (BWA)[53] version 0.7.13 was used to map data to the genome. Unmapped and low-quality ($q < 15$) reads were discarded. SAMtools[54] version 1.5 was used to remove duplicate sequences and HOMER[55] version 4.10.4 was used to call peaks using the findPeaks command with default parameters. The command "-style factor" was used for TFs and the command "-style histone" was used for histone modifications. Stage- and condition-matched input DNA controls were used as background when calling peaks. The BEDtools[56] version 2.26.0 suite of programs was used to perform genomic algebra operations. Tag directories were created for each replicate using HOMER. Directories from each replicate were then combined, and peaks were called from the combined replicates using HOMER. These peaks were then intersected with pancreatic enhancers, hepatic enhancers, or alveolar enhancers, respectively. Pearson correlations for the intersecting peaks were calculated between each pair of replicates using the command multiBamSummary from the deepTools2 package[57] version 3.1.3. Correlations are provided in Supplementary Table 5.

**RNA isolation and sequencing (RNA-seq) and qRT-PCR.** RNA was isolated from cell samples using the RNeasy® Micro Kit (Qiagen) according to the manufacturer instructions. For each cell stage and condition analyzed between 0.1 and 1 × 10$^6$ cells were collected for RNA extraction. For qRT-PCR, cDNA synthesis was first performed using the iScript™ cDNA Synthesis Kit (Bio-Rad) and 500 ng of isolated RNA per reaction. qRT-PCR reactions were performed in triplicate with 10 ng of template cDNA per reaction using a CFX96™ Real-Time PCR Detection System and the iQ™ SYBR® Green Supermix (Bio-Rad). PCR of the TATA-binding protein (TBP) coding sequence was used as an internal control and relative expression was quantified via double delta CT analysis. For RNA-seq, stranded, single-end sequencing libraries were constructed from isolated RNA using the TruSeq® Stranded mRNA Library Prep Kit (Illumina®) and library sequencing was performed on either a HiSeq 4000 System (Illumina®) or NovaSeq 6000 System (Illumina®) with single-end reads of either 50 or 75 base pairs (bp). Sequencing was performed by the UCSD IGM core research facility. A complete list of RT-qPCR primer sequences is provided in Supplementary Table 6.

**RNA-seq data analysis.** Reads were mapped to the human genome consensus build (hg19/GRCh37) using the Spliced Transcripts Alignment to a Reference (STAR) aligner version 2.4[58]. Normalized gene expression (fragments per kilobase per million mapped reads; FPKM) for each sequence file was determined using Cufflinks[59] version 2.2.1 with the parameters:–library-type fr-firststrand–max-bundle-frags 10000000. Differential gene expression was determined using DESeq2[60]. Adjusted $P$-values < 0.05 and fold change ≥ 2 were considered significant. For RNA-seq corresponding to cells at the HP stage, one replicate was generated. For all other RNA-seq experiments, replicates from two independent hESC differentiations were generated. Pearson correlations between bam files corresponding to each pair of replicates were calculated and are provided in Supplementary Table 7.

**Assay for transposase accessible chromatin sequencing (ATAC-seq).** ATAC-seq[61] was performed on approximately 50,000 nuclei. The samples were permeabilized in cold permabilization buffer (0.2% IGEPAL-CA630 (Sigma, Cat# I8896), 1 mM DTT (Sigma, Cat# D9779), Protease inhibitor (Roche, Cat# 05056489001), 5% BSA (Sigma, Cat# A7906) in PBS (Thermo Fisher Scientific, Cat# 10010-23) for 10 min on the rotator in the cold room and centrifuged for 5 min at 500 x $g$ at 4 °C. The pellet was resuspended in cold tagmentation buffer (33 mM Tris-acetate (pH = 7.8) (Thermo Fisher Scientific, Cat# BP-152), 66 mM K-acetate (Sigma, Cat# P5708), 11 mM Mg-acetate (Sigma, Cat# M2545), 16% DMF (EMD Millipore, Cat# DX1730) in Molecular biology water (Corning, Cat# 46000-CM)) and incubated with tagmentation enzyme (Illumina, Cat# FC-121-1030) at 37 °C for 30 min with shaking at 500 rpm. The tagmented DNA was purified using MinElute PCR purification kit (QIAGEN, Cat# 28004). Libraries were amplified using NEBNext High-Fidelity 2X PCR Master Mix (NEB, Cat# M0541) with primer extension at 72 °C for 5 min, denaturation at 98 °C for 30 s, followed by 8 cycles of denaturation at 98 °C for 10 s, annealing at 63 °C for 30 s and extension at 72 °C for 60 s. After the purification of amplified libraries using MinElute PCR purification kit (QIA-GEN, Cat# 28004), double size selection was performed using SPRIselect bead (Beckman Coulter, Cat# B23317) with 0.55X beads and 1.5X to sample volume. Finally, libraries were sequenced on HiSeq4000 (Paired-end 50 cycles, Illumina).

**ATAC-seq data analysis.** ATAC-seq reads were mapped to the human genome (hg19/GRCh37) using Burrows-Wheeler Aligner[53] (BWA) version 0.7.13, and visualized using the UCSC Genome Browser[52]. SAMtools[54] was used to remove unmapped, low-quality ($q < 15$), and duplicate reads. MACS2[62] version 2.1.4 was used to call peaks, with parameters "shift set to 100 bps, smoothing window of 200 bps" and with "nolambda" and "nomodel" flags on. MACS2 was also used to call ATAC-Seq summits, using the same parameters combined with the "call-summits" flag.

For all ATAC-seq experiments, replicates from two independent hESC differentiations were generated. Bam files for each pair of replicates were merged

for downstream analysis using SAMtools, and Pearson correlations between bam files for each individual replicate were calculated over a set of peaks called from the merged bam file. Correlations were performed using the command multiBamSummary from the deepTools2 package[57] with the "–removeOutliers" flag. Correlations are provided in Supplementary Table 8.

**Hi-C data analysis**. Hi-C data were processed as previously described[63]. Read pairs were aligned to the hg19 reference genome separately using BWA-MEM with default parameters[53]. Specifically, chimeric reads were processed to keep only the 5' position and reads with low mapping quality (<10) were filtered out. Read pairs were then matched, and Picard tools were then used to remove PCR duplicates. Bam files with alignments were further processed into text format as required by Juicebox tools[64]. Juicebox tools were then applied to generate Hi-C files containing normalized contact matrices. All downstream analysis was based on 10 Kb resolution KR-normalized matrices.

Chromatin loops were identified by comparing each pixel with its local background, as described previously[65] with some modifications. Specifically, only the donut region around the pixel was compared to model the expected count. Briefly, the KR-normalized contact matrices at 10 Kb resolution were used as input for loop calling. For each pixel, distance-corrected contact frequencies were calculated for each surrounding bin and the average of all surrounding bins. The expected counts were then transformed to raw counts by multiplying the counts with the raw-to-KR normalization factor. The probability of observing raw expected counts was calculated using Poisson distribution. All pixels with $P$-value < 0.01 and distance less than 10 Kb were selected as candidate pixels. Candidate pixels were then filtered to remove pixels without any neighboring candidate pixels since they were likely false positives. Finally, pixels within 20 Kb of each other were collapsed and only the most significant pixel was selected. The collapsed pixels with $P$-value < $1 \times 10^{-5}$ were used as the final list of chromatin loops.

**Single-cell RNA-sequencing library preparation**. Pancreatic progenitor cells at day 11 of differentiation were allowed to settle in microcentrifuge tubes and washed with PBS. Cell aggregates were incubated with Accutase® at 37 °C until a single-cell suspension was obtained. Cells were then resuspended in 1 mL ice-cold flow buffer comprised of 0.2% BSA in PBS and stained with propidium iodide (Sigma, Cat# P4170) to distinguish live cells. 500,000 live cells were collected using a FAC-SAria™ Fusion Flow Sorter, and 10,000 cells per sample were then loaded onto a 10X Chromium Controller and run using Next GEM Single-Cell 3' v3.1 reagents. Library preparation was performed according to manufacturer's instructions, and libraries were sequenced using a NovaSeq S4 (Paired-end 100 bp reads, Illumina).

**Single-cell RNA-sequencing data analysis**. Sequencing reads were processed using CellRanger[66] version 6.0.0, and matrices generated by CellRanger were imported into Seurat[67] version 3 for further processing. Doublet cells (>8000 total features for control cells and >6000 total features for motif optimized cells), low-coverage cells (<3000 total features for control cells and <2500 total features for motif optimized cells), and poor-quality cells (>10% mitochondrial reads for both conditions) were removed from further analysis. Each dataset was Log Normalized with a scale factor of 10,000 using the command "NormalizeData." Percentage of mitochondrial genes was regressed out of each dataset using the command "ScaleData." Integration anchors for each dataset were identified using "FindIntegrationAnchors," and datasets were integrated using the command "IntegrateData." Principal component analysis was performed for the integrated dataset using the command "RunPCA," and UMAP plots were generated through "RunUMAP." Clusters were defined running the commands "FindNeighbors" and "FindClusters" at a resolution of 0.03, and marker genes were identified using "FindMarkers." Feature plots and dot plots were generated using the commands "Featureplot" and "Dotplot," and differential expression of genes co-expressed with *NKX6.1* was calculated by subsetting for cells expressing *NKX6.1* and using "FindMarkers" to determine differential genes between control and motif optimized cells. Wilcoxon rank sum tests were used to calculate differential expression.

**Gene ontology analysis**. Gene ontology analysis for enhancer groups was performed using GREAT[68] version 4.0.4 with the default parameters. Gene ontology for differentially expressed genes and genes associated with class I and class II enhancers was performed using Metascape[69] using default parameters.

**Identification of super-enhancers**. To define pancreatic super-enhancers, we first identified pancreatic enhancers as distal genomic regions exhibiting a ≥ 2-fold increase in H3K27ac ChIP-seq signal during pancreas induction. We then used Rank Ordering of Super-enhancers (ROSE) software[21,70] to join identified pancreatic enhancers within a 12.5 kb span and rank these joined enhancers based on intensity of H3K27ac ChIP-seq signal. These joined enhancers were plotted based on H3K27ac signal, and pancreatic super-enhancers were defined as joined enhancers ranking above the inflection point of the resulting graph.

**Principal component analysis**. For RNA-seq data, transcriptomes were first filtered for genes expressed (FPKM ≥ 1) in at least one condition, then log10 transformed. For distal H3K27ac signals, H3K27ac peaks were filtered for distal enhancers (≥ 2.5 kb from any annotated TSS). Based on filtered values, PCA plots were generated using the PRComp package in R.

**Quantification of changes in H3K27ac signal**. HOMER[55] was used to annotate raw H3K27ac ChIP-seq reads over distal enhancers at developmental stages both before and after lineage induction. HOMER was then used to invoke the R package DESeq2[60] version 3.10 for differential analysis, using default parameters.

**Quantification of changes in TF ChIP-seq and ATAC-seq signal**. HOMER[55] was used to annotate raw FOXA1 and FOXA2 ChIP-seq reads, as well as ATAC-seq reads over PDX1-bound class I and class II enhancers in cells transfected with *SCRAM* and sh*PDX1* lentivirus. HOMER was then used to invoke the R package DESeq2[60] for differential analysis, using the flag "norm2total."

**Assignment of enhancer target genes**. RNA-seq data were filtered for expressed genes (FPKM ≥ 1) at the PP2 stage, and BEDTools[56] "closest" command was used to assign each enhancer to the nearest annotated TSS.

**Motif enrichment analysis**. HOMER[55] was used for comparative motif enrichment analyses, using the command findMotifsGenome.pl. de novo motifs were assigned to TFs based on suggestions generated by HOMER.

**Identification of FOXA motifs and generation of log-odds scores**. FOXA1 and FOXA2 PWMs were selected to encompass the most divergent PWMs for each TF. PWMs were downloaded from the JASPAR database[27], and occurrences with associated log-odds scores were quantified using the FIMO feature within the MEMEsuit package[71] version 5.1.1.

**Calculation of positional motif enrichment**. Identified ATAC-seq summits on class I and class II enhancers were flanked by 500 bp in each direction, and the CENTRIMO feature within the MEMEsuit package[72] version 5.1.1 was used to determine enrichment at summits for selected PWMs associated with FOXA1 and FOXA2, as well as to graph the positional probability of motif occurrence with respect to ATAC-seq summits.

**ATAC-seq footprinting analysis**. ATAC-seq footprinting was performed as previously described[73]. In brief, diploid genomes for CyT49 were created using vcf2diploid (version 0.2.6a)[74] and genotypes called from whole genome sequencing and scanned for a compiled database of TF sequence motifs from JASPAR[75] and ENCODE[76] with FIMO (version 4.12.0)[71] using default parameters for p-value threshold and a 40.9% GC content based on the hg19 human reference genome. Footprints within ATAC-seq peaks were discovered with CENTIPEDE (version 1.2)[77] using cut-site matrices containing Tn5 integration counts within a ± 100 bp window around each motif occurrence. Footprints were defined as those with a posterior probability ≥0.99.

**Permutation-based significance**. A random sampling approach (10,000 iterations) was used to obtain null distributions for enrichment analyses, in order to obtain $P$-values. Null distributions for enrichments were obtained by randomly shuffling enhancer regions using BEDTools[56] and overlapping with FOXA1/2-binding sites. $P$-values < 0.05 were considered significant.

**Quantification and statistical analysis**. Statistical analyses were performed using GraphPad Prism (v8.1.2), and R (v3.6.1). Statistical parameters such as the value of n, mean, standard deviation (SD), standard error of the mean (SEM), significance level (n.s., not significant; *$P$ < 0.05; **$P$ < 0.01; and ***$P$ < 0.001), and the statistical tests used are reported in the figures and figure legends. Unless otherwise noted, the "n" refers to the number of independent hESC differentiation experiments analyzed (biological replicates). All bar graphs and line graphs are displayed as mean ± S.E.M, and all box plots are centered on median, with box encompassing 25th–75th percentile and whiskers extending up to 1.5 interquartile range. Statistically significant gene expression changes were determined with DESeq2[60], and significantly enriched gene ontology terms were identified using Metascape[69].

For all bar graphs of gene expression measured via qPCR, each plotted point represents the average of three technical replicates. For all immunofluorescence, representative images are shown from n ≥ 2 independent differentiations. For all flow cytometry analyses, representative plots are shown from n = 3 independent differentiations.

**Reporting summary**. Further information on research design is available in the Nature Research Reporting Summary linked to this article.

## Data availability

The data that support this study are available from the corresponding author upon reasonable request. All mRNA-seq, ChIP-seq, and ATAC-seq datasets generated for this study have been deposited at GEO under the accession number GSE148368. The following datasets used in this study were obtained from the GEO and ArrayExpress repositories—RNA-seq: Pancreatic differentiation of CyT49 hESC line (E-MTAB-1086); ChIP-seq: H3K27ac in CyT49 hESC, DE, GT, PP1, PP2 (GSE54471 and GSE149148), H3K27ac in CyT49 PP2 SCRAM and PP2 sh*PDX1* (GSE54471), H3K4me1 in CyT49 GT and PP2 (GSE54471 and GSE149148), RXR in CyT49 PP1 (GSE104840), PDX1 in CyT49 PP2 (GSE54471 and GSE149148), HNF6 in CyT49 PP2 (GSE149148), SOX9 in CyT49 PP2 (GSE149148), FOXA1 in CyT49 PP2 (GSE149148), FOXA2 in CyT49 PP2 (GSE149148); ATAC-seq: CyT49 GT and PP2 (GSE149148). Hi-C datasets were generated as a component of the 4D Nucleome Project[78]. Datasets corresponding to the PP2 stages of differentiation can be found under accession number 4DNES0LVRKBM [https://data.4dnucleome.org/experiment-set-replicates/4DNESOLVRKBM/]. Source data are provided with this paper.

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

## Acknowledgements

We thank Ileana Matta for assistance with ATAC-seq assays and library preparations, and members of the Sander laboratory, Dr. Christopher W. Benner, Dr. Emma Farley, and Dr. Xin Sun for helpful discussions and critical reading of the manuscript. We acknowledge support of the UCSD Human Embryonic Stem Cell Core for cell sorting and K. Jepsen and the UCSD IGM Genomic Center (supported by P30 DK063491) for library preparation and sequencing. This work was supported by grant T32 GM008666 (R.J.G.), R01 DK068471 and R01 DK078803 (M.S.), U54 DK107977 (B.R.), the I.M. Rosenzweig Junior Investigator Award from the Pulmonary Fibrosis Foundation (K.D.A.), R01 HL095993, R01HL128172, U01TR001810, and N01 75N92020C00005 (D.N.K.).

## Author contributions

A.W. and M.S. conceived the project. R.J.G., A.W., and M.S. designed experiments. A.W., R.J.G., D.K.L., N.K.V., D.A.R., K.D.A., J.W., A.R., and S.K. performed experiments. A.W., R.J.G., N.K.V., Y.Q., and J.C. analyzed sequencing data. R.J.G., A.W., and M.S. interpreted data. R.J.G. and M.S. wrote the manuscript. B.R., K.J.G., D.N.K., and M.S. supervised all research.

## Competing interests

K.J.G. does consulting for Genentech. The authors declare no other competing interests.
