## [Peer Review File · Nature Communications]

REVIEWER COMMENTS

Reviewer #1 (Remarks to the Author):

This study used a tissue culture differentiation scheme to investigate enhancer activation by the pioneer factor FOXA during the course of differentiation towards the pancreatic lineage. The scheme first generates cells deemed to be at the primitive gut-tube stage followed by early and late pancreatic progenitor cells. The basic observation of the paper is that two subsets of FOXA1/2 recruitment by ChIPSeq are observed depending on whether recruitment is already present at the gut-tube stage and maintained at the PP2 in contrast to a larger subset that shows detectable recruitment only at the PP2 but not GT stage. At the PP2 stage, the FOXA-bound enhancers appear to be fully active as indicated by the presence of H3K27ac. The authors then conclude that the class 1 enhancers are primed at the GT stage to be activated at the PP2 stage. This class 1 subset that is already bound at the GT stage is found to be enriched in well conserved FOXA motifs, whereas the class 2 groups that is interpreted to be unprimed at the GT stage is enriched for motifs of other transcription factors involved in pancreatic differentiation. The authors then show that a large number of the class 2 enhancers are dependent on the pancreatic differentiation in factor PDX1 for strong FOXA binding, ATACSeq and H3K27ac signals. Occupancy at these sites thus appears to require cooperation with other factors that may not have pioneer activity. Finally, similar observations are made for enhancers targeted toward either liver or lung alveolar fates.

While the observations are interesting, their interpretation is not as straightforward as suggested by the authors. The primary observation that there are putative enhancer sites that appear to be occupied by FOXA alone and other subsets that require cooperation with other lineage restricted factors is not so novel as there are now many examples of cooperation between pioneer and non-pioneer factors. The authors appear surprised at the ratios of these two classes of enhancers as they had been expected more of the first compared to the second class, but it is difficult to appreciate the soundness of this expectation and interpretation. It is in fact difficult to clearly interpret the observations without better characterization of the mixture of differentiating cells present at the different stages of the differentiation scheme. It would not be surprising to find out that each stage of the scheme contains a somewhat heterogeneous mixture of cells in spite of the clear-cut label given to each of them. This cell heterogeneity may also in part explain the somewhat ambiguous "primed" vs "unprimed" status given to the class 1 and class 2 enhancers. Indeed, the authors rely on the strength of the ATACSeq signal for this interpretation despite the fact that the patterns of the active enhancer mark H3K4me1 is not supporting this terminology. Indeed, both the class 1 and class 2 enhancers are marked by H3K4me1 at the GT stage and therefore H3K4me1 is already present. For many authors, the presence of H3K4me1 is the landmark of primed enhancers and in this case even the class 2 enhancers of the present work would be labeled as primed (e.g. Mayran et al., Nat Comms. 2019). In this context, the class 1 enhancers in GT cells already exhibit some level of enhancer activity as indicated by the bimodal distribution of the H3K4me1 that is indicative of some nucleosome depletion. The marking of both classes of enhancers by some levels of the active mark H3K4me1 is not surprising given that FOXA factors are already expressed at the DE earlier stage. Also, one cannot avoid noticing that the ChIPSeq signals for the FOXA factors are relatively weak: it is thus possible that these low signals account for the apparent absence of FOXA at the GT stage: is the difference truly black and white or could it be more of a quantitative difference? An altogether different interpretation of the results might be that for the class 2 enhancers the cooperating non-pioneer factor is only expressed in the PP stages (as shown in Fig S4C) and therefore available at these stages for stable binding with FOXA factors whereas for the class one enhancers, a cooperating factor is either already present or not essential. In such context, the novelty of the present observations is not so great and this raises questions about their biological relevance. To sum up those queries, it seems essential for the authors to better characterize the biological system used for these studies in order to support conclusions that are truly novel rather than reflecting the particularities of the in vitro differentiation schemes. The work is thus relatively premature and should be investigated further in

order to present a novel and well-supported conclusion.

Some specific queries:

1. On L180. The authors indicate that "class 2 enhancers acquire these features largely with pancreatic lineage induction". The qualificative "largely" indicates that the authors themselves recognise that the H3K4me1 mark patterns are not that clear-cut to define "primed vs unprimed" enhancers. Hence, this reviewer challenges this interpretation and the appropriateness of calling one group of enhancers primed but not the other. Both classes appear to be primed to some extent based on the H3K4me1 mark and therefore the difference between the two classes may just be levels of subsequent activation as reflected by the ATAC signals and shape of the H3K4me1 metaplots.
2. P.13 L278. The authors interpret the loss of FOXA binding after knockdown of PDX1 as occurring "to a greater extent at class 2 than class 1 enhancers (Fig.5B)". The differences are apparent but weak: some statistical evaluation of these differences might be warranted.
3. P.16 L326. The authors note that for the liver directed differentiation scheme FOXA motifs are enriched at class 1 enhancers: they neglect to mention that GATA motifs are as enriched as FOXA! Since both FOXA and GATA factors have pioneer activity, it is possible that cooperation between these two factors might account for the class 1-type behaviour of these enhancers. In relation to the above discussion about factor cooperation, this pair is a very likely pair of accomplices. This may merit discussion!

Reviewer #2 (Remarks to the Author):

This is an interesting study from the Sander lab. on enhancer engagement and regulation by FOXA transcription factors in human endoderm development. The authors examined enhancer activation (primed and unprimed) and mapped FOXA1 and FOXA2 binding using a combination of ATACSeq and ChiP-Seq in human ESC cultures undergoing pancreatic cell differentiation. Moreover, they determined the requirement of FOXA1/2 in pancreatic lineage induction using a CRISPR-mediated genetic approach. Overall, their findings support a stepwise mechanism of action for FOXA 1/2 at endodermal enhancers: first, a priming role in a small subset of enhancers, before lineage induction; second, activation of a larger set of enhancers through cooperation with lineage-specific transcription factors. Beside the pancreatic lineage, they expanded these observations to two close endodermal lineages, liver and lung.

The study is very well done and perhaps among the most mechanistic and in-depth analysis of enhancer activation and FOXA TF activity in human endoderm development to date. This work is important and raises intriguing questions about enhancer regulation and lineage plasticity within endodermal derivatives that might be relevant also in the context of cancer biology.

Minor points that could be addressed:

- FOXA1/2 mutant PPs are significantly different from the control counterpart; this is very visible in the PCA analysis in Fig. 1. Which identity the cells acquired in the absence of FOXA1/2?
- The concept of super-enhancer is very interesting and relatively unexplored in this context. What were criteria used here for defining 167 super-enhancers among the identified pancreatic enhancers?

Reviewer #3 (Remarks to the Author):

In this study Geusz et al. use human pluripotent stem cell (hPSC) differentiation into foregut endodermal lineages as a model to study the role of pioneer transcription factors FOXA1/2 in lineage induction. The chief finding is that there are two modes of binding of the FOX proteins to enhancers: one that precedes lineage induction and is associated with enhancer priming, and a second that engages enhancers on lineage induction and requires lineage specific transcription factors. The authors first establish the requirement for FOXA2 in pancreatic induction (one which can be partially rescued by FOXA1), but not in early endoderm specification. The authors then carried out a time course study of enhancer binding during induction of pancreatic specification, revealing a biphasic pattern of enhancer binding. The first set of enhancer binding occurred in the early stage and was associated with enhanced chromatin accessibility, whereas the second set, representing the majority of bound enhancers, were bound on lineage induction, were associated with binding sites for pancreatic lineage specific factors, and were PDX1 dependent. A similar biphasic pattern was observed in liver and lung differentiation paradigms.

The study reports extensive data and global chromatin analysis to support these conclusions and the work is clearly presented. It leaves some questions unanswered particularly in respect of the role of FOXA1/2 priming. The role of binding to Class 1 enhancers is unclear. A third class of enhancers specific to GT stage is not analyzed. Perhaps conditional knockouts or more in depth analysis of wild type and FOXA1/2 KO at the gut tube stage would shed light on these questions. The end of the Discussion seems to struggle a bit in an effort to reach some explanations. There are some specific points the authors could address to help elucidate some of these outstanding mysteries.

Specific points

1. P 5-6 L 111 What about DE or GT stage-what is the evidence that the double KO cells are equivalent to wild type? Expression of two markers does not prove this. The WT and KO are somewhat separated in PC1 in Figure 1g and clearly in PC2 in Supp. 2C.
2. L 156. What is the role of Class 3 Enhancers-what are they doing at GT stage? Just because they are a minority does not mean they do not have significant function. Also they seem to have higher H3K27ac at PP2 in Figure 2b.
3. L 159 is there a "bookmarking" function of Class 1 enhancers?
4. L 188 how are superenhancers defined/identified here
5. Fig 3e more in depth analysis of the class of genes associated with Class 1 and 2 enhancers would be informative
6. L216 Fig 4a are these the only enriched motifs? What determined the cutoff for display in this figure?
7. L 233 4d explain terminology here for those unfamiliar with this field, what do MAO refer to? How were these singled out for analysis?
8. L 316 and Figure 6d FOXA1 signal increased at class1 in ALV stage, apparent contrast to pancreas and liver?
9. Figure 6 g,h-similar question to 6 above.
10. L 414 function of primed enhancers remains unclear, what do the authors think pioneering activity really means?
11. L 442 onward-this is somewhat confusing and speculative.
12. L 457 why only a subset of primed hepatic enhancers lost after pancreas induction. What is this subset?

Revision overview

We thank the reviewers for their constructive and thoughtful comments on the manuscript and appreciate that the reviewers found the experiments well-performed and the data clear and valuable. However, reviewers 1 and 3 also noted a need for additional analysis of the role of FOXA transcription factors (TFs) in enhancer priming, as well as a more detailed examination of the biological relevance of class I and class II enhancers. In addition, reviewer 1 commented that potential heterogeneity in our cell populations could obscure correct interpretation of our findings. We addressed these shortcomings with additional experiments. Major novel data and findings include:

1. **Demonstration that altering FOXA motif strength is sufficient to redefine temporal patterns of FOXA recruitment during development.** To substantiate our main conclusion that regulatory DNA sequence logic governs temporal FOXA recruitment during organ lineage development, we optimized FOXA binding motif strength in an enhancer near the pancreatic beta cell lineage-determining TF *NKX6.1* via gene editing in hESCs. We show that optimizing FOXA binding motifs redefines temporal patterns of FOXA binding and H3K4me1 deposition and obviates the requirement of lineage-specific TFs for FOXA recruitment (**Updated Fig. 6a-d**). These findings directly link developmental enhancer priming to DNA sequence logic.
2. **Evidence that altering FOXA motif strength affects the threshold for target gene expression.** Using single cell RNA-seq analysis, we show that FOXA motif optimization lowers the target gene expression threshold and renders gene expression independent of high levels of lineage-specific TFs (**Updated Fig. 6c,d** and **Updated Supp. Fig. 7a-e**). Thus, by resolving heterogeneity in our cell populations through single cell analysis, we causally link mechanisms of FOXA binding to target gene expression thresholds. These findings substantiate our conclusion that class II enhancers impose a lineage-specific barrier to gene expression by coupling FOXA occupancy to cooperative binding with lineage-specific TFs.
3. **Demonstration that altering FOXA motif strength in a single enhancer is sufficient to change cell fate.** Since FOXA motif optimization in the *NKX6.1* enhancer broadened the domain of *NKX6.1* expression among pancreatic progenitors, we examined cell fate choices in motif optimized and control cells. Consistent with *NKX6.1*'s known role as a beta cell fate determinant, motif optimization evoked an alpha-to-beta cell fate switch (**Updated Fig. 6e** and **Updated Supp. Fig. 7f,g**), demonstrating the biological relevance of FOXA binding mechanisms at developmental enhancers for cell type specification.
4. **Evidence for class I enhancers being signal-responsive.** Additional characterization of class I enhancers revealed binding of GATA TFs (**Updated Fig. 4b** and **Updated Supp. Fig. 9f**) which are known to cooperate with FOXA TFs. Furthermore, we show binding of the signal-dependent TF RXR (**Updated Fig. 4b**) and comparative enrichment in class I enhancer-associated genes for components of signal transduction pathways (**Updated Supp. Fig. 4g**), supporting that class I enhancers are signal-responsive.

Together, our new data substantiate our model whereby early FOXA binding to class I enhancers confers responsiveness to lineage-inductive signals in multipotent intermediates, while class II enhancers restrict target gene expression to a specific lineage. Our new findings support the concept that FOXA TF motif strength at developmental enhancers provides a tunable threshold for target gene expression.

Response to reviewers' comments

Reviewer #1

This study used a tissue culture differentiation scheme to investigate enhancer activation by the pioneer factor FOXA during the course of differentiation towards the pancreatic lineage. The scheme first generates cells deemed to be at the primitive gut-tube stage followed by early and late pancreatic progenitor cells. The basic observation of the paper is that two subsets of FOXA1/2 recruitment by ChIPSeq are observed depending on whether recruitment is already present at the gut-tube stage and maintained at the PP2 in contrast to a larger subset that shows detectable recruitment only at the PP2 but not GT stage. At the PP2 stage, the FOXA-bound enhancers appear to be fully active as indicated by the presence of H3K27ac. The authors then conclude that the class 1 enhancers are primed at the GT stage to be activated at the PP2 stage. This class 1 subset that is already bound at the GT stage is found to be enriched in well conserved FOXA motifs, whereas the class 2 groups that is interpreted to be unprimed at the GT stage is enriched for motifs of other transcription factors involved in pancreatic differentiation. The authors then show that a large number of the class 2 enhancers are dependent on the pancreatic differentiation in factor PDX1 for strong FOXA binding, ATACSeq and H3K27ac signals. Occupancy at these sites thus appears to require cooperation with other factors that may not have pioneer activity. Finally, similar observations are made for enhancers targeted toward either liver or lung alveolar fates.

While the observations are interesting, their interpretation is not as straightforward as suggested by the authors. The primary observation that there are putative enhancer sites that appear to be occupied by FOXA alone and other subsets that require cooperation with other lineage restricted factors is not so novel as there are now many examples of cooperation between pioneer and non-pioneer factors. The authors appear surprised at the ratios of these two classes of enhancers as they had been expected more of the first compared to the second class, but it is difficult to appreciate the soundness of this expectation and interpretation. It is in fact difficult to clearly interpret the observations without better characterization of the mixture of differentiating cells present at the different stages of the differentiation scheme. It would not be surprising to find out that each stage of the scheme contains a somewhat heterogeneous mixture of cells in spite of the clear-cut label given to each of them. This cell heterogeneity may also in part explain the somewhat ambiguous “primed” vs “unprimed” status given to the class 1 and class 2 enhancers. Indeed, the authors rely on the strength of the ATACSeq signal for this interpretation despite the fact that the patterns of the active enhancer mark H3K4me1 is not supporting this terminology. Indeed, both the class 1 and class 2 enhancers are marked by H3K4me1 at the GT stage and therefore H3K4me1 is already present. For many authors, the presence of H3K4me1 is the landmark of primed enhancers and in this case even the class 2 enhancers of the present work would be labeled as primed (e.g. Mayran et al., Nat Comms. 2019). In this context, the class 1 enhancers in GT cells already exhibit some level of enhancer activity as indicated by the bimodal distribution of the H3K4me1 that is indicative of some nucleosome depletion. The marking of both classes of enhancers by some levels of the active mark H3K4me1

is not surprising given that FOXA factors are already expressed at the DE earlier stage. Also, one cannot avoid noticing that the ChIPSeq signals for the FOXA factors are relatively weak: it is thus possible that these low signals account for the apparent absence of FOXA at the GT stage: is the difference truly black and white or could it be more of a quantitative difference? An altogether different interpretation of the results might be that for the class 2 enhancers the cooperating non-pioneer factor is only expressed in the PP stages (as shown in Fig S4C) and therefore available at these stages for stable binding with FOXA factors whereas for the class one enhancers, a cooperating factor is either already present or not essential. In such context, the novelty of the present observations is not so great and this raises questions about their biological relevance. To sum up those queries, it seems essential for the authors to better characterize the biological system used for these studies in order to support conclusions that are truly novel rather than reflecting the particularities of the in vitro differentiation schemes. The work is thus relatively premature and should be investigated further in order to present a novel and well-supported conclusion.

We are grateful for the thorough evaluation by this reviewer and his/her acknowledgement that our data is of interest. We have performed several experiments and analyses to address the concerns raised. Before addressing each point in detail, we here summarize the most significant revision which is the experimental demonstration that FOXA motif strength dictates temporal patterns of FOXA recruitment during organ lineage specification. These new findings are shown in **Updated Fig. 6** and **Updated Supp. Fig. 7**.

We used gene editing in hESCs to “strengthen” weak FOXA binding motifs at an unprimed class II enhancer near the pancreatic beta cell lineage-determining TF *NKX6.1* and differentiated edited and control cells to the PP2 pancreatic progenitor cell stage. This enhancer lacks FOXA binding (**Updated Fig. 2d – was Fig. 2c in original submission**), accessible chromatin (**Supp. Fig. 4b**), and H3K4me1 signal (**Supp. Fig. 4b**) prior to pancreas induction. Furthermore, in the absence of PDX1, FOXA does not bind to the enhancer and enhancer chromatin does not gain accessibility or activation marks (**Fig. 5c**). Consistent with our hypothesis that FOXA motif strength determines temporal FOXA binding, optimizing FOXA motifs resulted in FOXA recruitment to the *NKX6.1* enhancer prior to pancreatic lineage induction. This early recruitment of FOXA led to deposition of H3K4me1 prior to enhancer activation, showing that FOXA mediates H3K4me1 deposition without activating the enhancer. Thus, we show that optimization of FOXA binding motifs can uncouple FOXA recruitment from cooperative interactions with PDX1 and convert an unprimed enhancer into a primed enhancer.

Given potential cell heterogeneity at the PP2 stage, we further conducted single cell RNA-seq analysis of PP2 stage cells from control and motif optimized cell lines to determine how FOXA motif strength affects gene expression and PP2 stage cell type composition. While *NKX6.1* expression in control PP2 stage cells was restricted to a subpopulation of cells expressing high levels of *PDX1*, *NKX6.1* was more broadly expressed in motif optimized cultures, including in cells expressing lower levels of *PDX1*. These findings indicate that optimizing FOXA motif strength lowers the threshold

for enhancer activation and *NKX6.1* target gene expression, which we confirmed at the protein level. Thus, we show that increasing FOXA motif strength renders *NKX6.1* expression independent of high levels of PDX1, which corroborates our model and conclusions. Given that alpha cells are derived from *NKX6.1*⁻ endocrine progenitors, whereas beta cells arise from *NKX6.1*⁺ endocrine progenitors¹, we further examined effects of ectopic expression of *NKX6.1* on cell fate allocation and observed a pre-alpha to a pre-beta cell fate shift. Therefore, our results provide evidence for the importance of FOXA binding mechanisms at developmental enhancers for cell type specification.

These new data not only validate the model proposed in our initial submission but substantially expand insights by showing that changes in FOXA binding motif strength (i) redefine patterns of FOXA recruitment during development, (ii) uncouple FOXA binding from lineage-determining TFs, (iii) alter the threshold for target gene expression, and (iv) change cell fate choice. While other studies have reported a correlation between FOXA motif strength and the ability of FOXA to displace nucleosomes²⁻⁴, our results are the first to directly demonstrate that motif strength is a determinant of FOXA binding affinity *in vivo* and that motif strength helps fine-tune developmental gene regulation with relevance to organ cell type composition.

It would not be surprising to find out that each stage of the scheme contains a somewhat heterogeneous mixture of cells in spite of the clear-cut label given to each of them. This cell heterogeneity may also in part explain the somewhat ambiguous “primed” vs “unprimed” status given to the class 1 and class 2 enhancers.

We thank the reviewer for the helpful suggestion of probing heterogeneity within our system. As detailed above, we profiled PP2 stage cells by scRNA-seq and found that PP2 cells cluster in two different populations, one with high and one with low expression of pre-endocrine genes (**Updated Fig. 6c** and **Updated Supp. Fig. 7a**). We agree that cell heterogeneity might introduce some ambiguity in assigning class I and class II status to enhancers. However, our motif optimization experiment clearly shows that early FOXA recruitment and enhancer priming are dependent on FOXA binding motif sequence (**Updated Fig. 6b**). Therefore, despite some ambiguity in assigning class I or class II status to enhancers from bulk data, our new findings validate that there are two classes of enhancers and that the ability of FOXA to engage chromatin before lineage induction and to induce H3K4me1 deposition depends on FOXA motif strength.

Indeed, the authors rely on the strength of the ATACSeq signal for this interpretation despite the fact that the patterns of the active enhancer mark H3K4me1 is not supporting this terminology. Indeed, both the class 1 and class 2 enhancers are marked by H3K4me1 at the GT stage and therefore H3K4me1 is already present. For many authors, the presence of H3K4me1 is the landmark of primed enhancers and in this case even the class 2 enhancers of the present work would be labeled as primed (e.g. Mayran et al., Nat Comms. 2019). In this context, the class 1 enhancers in GT cells already exhibit some level of enhancer activity as indicated by the bimodal distribution of the H3K4me1 that is indicative of some nucleosome depletion. The marking of both

classes of enhancers by some levels of the active mark H3K4me1 is not surprising given that FOXA factors are already expressed at the DE earlier stage.

We agree that H3K4me1 is an important marker of primed enhancers. To this end, we quantified class I and class II enhancers overlapping regions of H3K4me1 at the GT stage. Within a 1 kb span, we find that 62% of class I enhancers compared to 34% of class II enhancers overlap regions of H3K4me1. Thus, while a minor population of class II enhancers may be primed, this is not the predominant pattern within this group of enhancers. We added this data as **Supp. Fig. 4c** and included a qualifier in the results section to acknowledge that there is a small population of class II enhancers with H3K4me1 deposition at the GT stage.

(L193): “Although a subset of class II enhancers was marked by H3K4me1 at the GT stage, this population comprised the minority of class II enhancers (**Supplementary Fig. 4c**).”

We most directly address the question of enhancer priming through our experiments in **Updated Fig. 6b**, where we show that recruitment of FOXA TFs to an edited class II enhancer leads to the accumulation of H3K4me1 signal in the absence of H3K27ac signal. This is the *bona fide* definition of a primed enhancer⁵. Thus, we are confident in saying that, by and large, the presence of FOXA TFs prior to pancreas induction at class I enhancers confers enhancer priming, while the absence of FOXA TFs at class II enhancers is associated with an unprimed state.

Furthermore, while we agree that the bimodal distribution of H3K4me1 at class I enhancers is indicative of nucleosome depletion, we emphasize that chromatin accessibility does not imply enhancer activation. Indeed, accessible chromatin is a known feature of primed enhancers⁶. Lastly, to address the concern that FOXA TFs could prime class II enhancers at the DE stage, we performed ChIP-seq for FOXA1 and FOXA2 in DE stage cells, as shown in **Reviewer 1 Figure 1**. We show that at the DE stage, only 4% of class II enhancers show association with FOXA TFs. Thus, we deem it unlikely that FOXA TFs prime class II enhancers at the DE stage. At the reviewer’s request, we will incorporate these data into the manuscript.

Reviewer 1 Figure 1. Class I and class II enhancers show little overlap with FOXA peaks at the DE stage. Pie charts showing overlap of class I and class II enhancers with FOXA1 or FOXA2 ChIP-seq binding sites at the DE stage.

Also, one cannot avoid noticing that the ChIPSeq signals for the FOXA factors are relatively weak: it is thus possible that these low signals account for the apparent absence of FOXA at the GT stage: is the difference truly black and white or could it be more of a quantitative difference?

Regarding FOXA1 and FOXA2 ChIP-seq signal at the GT stage, we note that for each transcription factor ChIP-seq experiment we generated 2 independent biological replicates which yield highly correlated sets of peaks (Pearson=0.90, data shown in “ChIP-seq data analysis” portion of Methods section). The antibodies have been previously used for ChIP-seq in several studies⁷⁻¹², and all datasets meet quality benchmarks set by the ENCODE consortium (https://www.encodeproject.org/chip-seq/transcription_factor/). To address the possibility that the lack of FOXA signal at class II enhancers at the GT stage results from overall low ChIP-seq signal, we performed two analyses:

- 1) We called peaks in our ChIP-seq datasets at a much lower threshold (5% false discovery rate compared to 0.1% used in the manuscript). This increased sensitivity may lead to false positives but should better distinguish low-level peaks that may otherwise be missed. However, even using this low threshold, we find that less than 1% of class II enhancers exhibit a FOXA peak at the GT stage, compared to 100% of class I enhancers.
- 2) We downloaded published FOXA2 ChIP-seq data from GT stage cells generated in an analogous differentiation system¹³. Intersection of class I and class II enhancers with this independent dataset revealed that 79% of class I enhancers are FOXA2-bound, whereas only 18% of class II enhancers are bound by FOXA2.

Based on this evidence, we are confident in our conclusion that lack of FOXA peaks overlapping class II enhancers prior to pancreas induction is the result of biological rather than technical contributors.

An altogether different interpretation of the results might be that for the class 2 enhancers the cooperating non-pioneer factor is only expressed in the PP stages (as shown in Fig S4C) and therefore available at these stages for stable binding with FOXA factors whereas for the class one enhancers, a cooperating factor is either already present or not essential. In such context, the novelty of the present observations is not so great and this raises questions about their biological relevance.

The primary goal of our study was not to identify novel mechanisms of FOXA recruitment to DNA but rather to determine *in vivo* relevance of biochemically identified mechanisms of FOXA recruitment for developmental gene regulation and cell lineage decisions. Towards this goal, we demonstrate that enhancer activation during endodermal lineage induction is dependent upon FOXA TFs, and that two classes of enhancers can be distinguished based on differences in temporal FOXA TF recruitment and DNA sequence logic. The new results shown in **Updated Fig. 6** further show that DNA sequence is indeed the driver for temporal differences in FOXA recruitment and that motif strength affects the target gene expression threshold to influence cell lineage

decisions. Thus, by showing that motif strength helps fine-tune developmental gene regulation our new results demonstrate biological relevance.

Furthermore, to study biological differences between the two classes of enhancers, we determined whether binding of signal-dependent TFs is enriched at class I enhancers and whether genes associated with class I enhancers are functionally different from class II-associated genes. Consistent with the enrichment of binding motifs for signal-dependent TFs at class I enhancers (**Updated Fig. 4a**), we found that class I enhancers are preferentially bound by the retinoic acid receptor subunit RXR (**Updated Fig. 4b**), suggesting that class I enhancers are more likely to interpret lineage-inductive signals than class II enhancers. Furthermore, comparison of class I with class II enhancer-associated genes revealed involvement of class I enhancer-associated genes in intracellular signal transduction (**Updated Supp. Fig. 4g**). Thus, lineage-inductive signals (e.g., retinoic acid for pancreas) are likely interpreted at class I enhancers to modulate intracellular signaling events in lineage intermediates. These findings show differences between the gene regulatory programs controlled by class I and class II enhancers, linking gene regulatory mechanisms to biological function. A detailed discussion of the biological relevance of our findings is included from line 509-551.

It is possible that other TFs could act cooperatively with FOXA TFs at class I enhancers. Such cooperativity would not contradict our model or conclusions. Since GATA motifs are enriched at class I enhancers (**Updated Fig. 4a**), we determined whether GATA4 and GATA6 bind to class I enhancers at the GT stage, when GATA4 and GATA6 are already expressed. Indeed, we found that close to half of the class I enhancers bound GATA4 or GATA6 at the GT stage, whereas a very small percentage of class II enhancers bound GATA4 or GATA6 (**Updated Fig. 4b**). Irrespective of potential cooperativity with other TFs at class I enhancers, our new data identify FOXA motif strength as the primary determinant of whether FOXA TFs can engage with DNA prior to lineage induction independent of lineage-specific TFs. Since GATA TFs do not bind to the *NKX6.1* enhancer, our data in motif optimized cell lines show that early FOXA recruitment can occur independent of GATA TFs when low affinity FOXA binding motifs are converted into high affinity motifs.

We added the following paragraph to the discussion:

L498-508: “*Our observation that FOXA1/2 bind primed enhancers without cooperative recruitment by pancreatic TFs raises the question of how FOXA TFs engage their target sites at primed enhancers. We find that a subset of primed enhancers is bound by both FOXA and GATA TFs prior to lineage induction. Given previously demonstrated cooperativity between FOXA and GATA TFs¹⁷, it is possible that GATA TFs help recruit FOXA to a subset of primed enhancers. However, we show that strengthening FOXA motifs is sufficient to enable FOXA1/2 binding to an enhancer not bound by GATA TFs. Therefore, our data support the conclusion that strong FOXA motifs are sufficient to facilitate FOXA TF engagement and chromatin priming during development, consistent with observations that FOXA1/2 can engage target sites on nucleosomal DNA in vitro⁷⁻⁹.*”

To sum up those queries, it seems essential for the authors to better characterize the biological system used for these studies in order to support conclusions that are truly novel rather than reflecting the particularities of the in vitro differentiation schemes. The work is thus relatively premature and should be investigated further in order to present a novel and well-supported conclusion.

We are confident that the more thorough characterization of our biological system through additional ChIP-seq experiments (**Updated Fig. 4b**), single cell RNA-seq analysis (**Updated Fig. 6** and **Updated Supp. Fig. 7**), and perturbation experiments (**Updated Fig. 6** and **Updated Supp. Fig. 7**) offer validation of and additional insight into the conclusions reached in our initial submission. Our new data establish that organ lineage gene regulatory programs are established by two classes of enhancers distinguished by mechanisms of FOXA TF recruitment and driven by FOXA motif strength. Mechanistically, we show that FOXA motif strength is linked to target gene expression thresholds, which is an entirely novel finding.

Some specific queries:

1. On L180. The authors indicate that “class 2 enhancers acquire these features largely with pancreatic lineage induction”. The qualitative “largely” indicates that the authors themselves recognise that the H3K4me1 mark patterns are not that clear-cut to define “primed vs unprimed” enhancers. Hence, this reviewer challenges this interpretation and the appropriateness of calling one group of enhancers primed but not the other. Both classes appear to be primed to some extent based on the H3K4me1 mark and therefore the difference between the two classes may just be levels of subsequent activation as reflected by the ATAC signals and shape of the H3K4me1 metaplots.

Please refer to our earlier comments on H3K4me1 and **Reviewer 1 Figure 1**. As noted, we added qualifying statements to address the small group of class II enhancers that show overlap with H3K4me1 signal.

2. P.13 L278. The authors interpret the loss of FOXA binding after knockdown of PDX1 as occurring “to a greater extent at class 2 than class 1 enhancers (Fig.5B)”. The differences are apparent but weak: some statistical evaluation of these differences might be warranted.

We quantified more rigorously the extent to which class I and class II enhancers lose FOXA binding in the absence of PDX1. In **Updated Supp. Fig. 6b**, we show that 23% of PDX1-bound class II enhancers show a significant reduction in FOXA ChIP-seq signal (≥ 2 -fold change, $P. \text{adj.} < 0.05$), while only 3% of PDX1-bound class I enhancers show this reduction. Additionally, we quantified the degree to which FOXA ChIP-seq signal changes at PDX1-bound class I and class II enhancers. We observed that loss of FOXA signal is significantly greater at class II enhancers than at class I enhancers (**Updated Supp. Fig 6c**).

3. P.16 L326. The authors note that for the liver directed differentiation scheme FOXA motifs are enriched at class 1 enhancers: they neglect to mention that GATA motifs are as enriched as FOXA! Since both FOXA and GATA factors have pioneer activity, it is possible that cooperation between these two factors might account for the class 1-type behaviour of these enhancers. In relation to the above discussion about factor cooperation, this pair is a very likely pair of accomplices. This may merit discussion!

As discussed above, we agree that FOXA and GATA TFs could cooperate at class I enhancers. This possibility was also discussed in the Discussion section of our original submission (L428-L431). Similar to pancreatic class I enhancers at the GT stage (**Updated Fig. 4b**), hepatic class I enhancers were also bound by GATA4 and GATA6 at the GT stage to a larger extent than hepatic class II enhancers. We show these data in **Updated Supp. Fig. 9f** and updated the Discussion section (L498-508) as detailed above.

Reviewer #2

This is an interesting study from the Sander lab. on enhancer engagement and regulation by FOXA transcription factors in human endoderm development. The authors examined enhancer activation (primed and unprimed) and mapped FOXA1 and FOXA2 binding using a combination of ATACSeq and ChiP-Seq in human ESC cultures undergoing pancreatic cell differentiation. Moreover, they determined the requirement of FOXA1/2 in pancreatic lineage induction using a CRISPR-mediated genetic approach. Overall, their findings support a stepwise mechanism of action for FOXA 1/2 at endodermal enhancers: first, a priming role in a small subset of enhancers, before lineage induction; second, activation of a larger set of enhancers through cooperation with lineage-specific transcription factors. Beside the pancreatic lineage, they expanded these observations to two close endodermal lineages, liver and lung.

The study is very well done and perhaps among the most mechanistic and in-depth analysis of enhancer activation and FOXA TF activity in human endoderm development to date. This work is important and raises intriguing questions about enhancer regulation and lineage plasticity within endodermal derivatives that might relevant also in the context of cancer biology.

We appreciate the recognition of the quality of our work and the advances it provides. As detailed in our revision overview, we further strengthened our conclusions by more clearly defining the role of class I enhancers and by demonstrating that altering FOXA motif strength changes patterns of FOXA recruitment, dependency of FOXA binding on other TFs, chromatin priming, and the threshold for target gene expression. These new data are shown in **Updated Fig. 6** and **Updated Supp. Fig. 7**.

Below we address the points raised by this reviewer.

Minor points that could be addressed:

FOXA1/2 mutant PPs are significantly different from the control counterpart; this is very visible in the PCA analysis in Fig. 1. Which identity the cells acquired in the absence of FOXA1/2?

To address this question, we identified 3124 genes that are upregulated in FOXA1/2 mutant cells compared to control cells at the PP2 stage (≥ 2 -fold decrease, P . adj. < 0.05) and performed gene ontology analysis. We observed an upregulation of genes associated with processes such as heart and skeletal system development (**Reviewer 2 Figure 1**). This suggests that in addition to activating endodermal genes, FOXA TFs may suppress genes associated with alternate germ layers through either direct or indirect mechanisms. Although this is an intriguing possibility, we believe that this observation would need to be substantiated with additional experiments. While interesting, such experiments would take the manuscript into a different direction. Therefore, we feel that this question is better left for future studies.

Reviewer 2 Figure 1. Upregulated genes in *FOXA1/2*^{-/-} PP2 cells compared to control cells. Enriched gene ontology terms of 3124 upregulated genes (≥ 2 -fold decrease, P adj. < 0.05) in *FOXA1/2*^{-/-} compared to control PP2 cells.

The concept of super-enhancer is very interesting and relatively unexplored in this context. What were criteria used here for defining 167 super-enhancers among the identified pancreatic enhancers?

To define pancreatic super-enhancers, we first identified pancreatic enhancers as distal genomic regions exhibiting a ≥ 2 -fold increase in H3K27ac ChIP-seq signal during pancreas induction. We then used Rank Ordering of Super-enhancers (ROSE) software^{14,15} to join identified pancreatic enhancers within a 12.5 kb span and rank these joined enhancers based on intensity of H3K27ac ChIP-seq signal. These joined enhancers were plotted based on H3K27ac signal as in **Supp. Fig. 4d**, and pancreatic super-enhancers were defined as joined enhancers ranking above the inflection point of the resulting graph. We have updated the Methods section (L967-975) of our resubmitted manuscript to make the identification process clearer.

Reviewer #3:

In this study Geusz et al. use human pluripotent stem cell (hPSC) differentiation into foregut endodermal lineages as a model to study the role of pioneer transcription factors FOXA1/2 in lineage induction. The chief finding is that there are two modes of binding of the FOX proteins to enhancers: one that precedes lineage induction and is associated with enhancer priming, and a second that engages enhancers on lineage induction and requires lineage specific transcription factors. The authors first establish the requirement for FOXA2 in pancreatic induction (one which can be partially rescued by FOXA1), but not in early endoderm specification. The authors then carried out a time course study of enhancer binding during induction of pancreatic specification, revealing a biphasic pattern of enhancer binding. The first set of enhancer binding occurred in the early stage and was associated with enhanced chromatin accessibility, whereas the second set, representing the majority of bound enhancers, were bound on lineage induction, were associated with binding sites for pancreatic lineage specific factors, and were PDX1 dependent. A similar biphasic pattern was observed in liver and lung differentiation paradigms.

The study reports extensive data and global chromatin analysis to support these conclusions and the work is clearly presented. It leaves some questions unanswered particularly in respect of the role of FOXA1/2 priming. The role of binding to Class 1 enhancers is unclear. A third class of enhancers specific to GT stage is not analyzed. Perhaps conditional knockouts or more in depth analysis of wild type and FOXA1/2 KO at the gut tube stage would shed light on these questions. The end of the Discussion seems to struggle a bit in an effort to reach some explanations. There are some specific points the authors could address to help elucidate some of these outstanding mysteries.

We are grateful for this reviewer's careful examination of our work, as well as the recognition of the extensive data we provide and the clarity of our manuscript. As detailed in the revision overview, we conducted extensive additional experimentation to investigate the role of FOXA TFs in enhancer priming more thoroughly. These new findings are shown in **Updated Fig. 6** and **Updated Supp. Fig. 7**.

We used gene editing in hESCs to "strengthen" weak FOXA binding motifs at an unprimed class II enhancer near the pancreatic beta cell lineage-determining TF *NKX6.1* and differentiated edited and control cells to the PP2 pancreatic progenitor cell stage. This enhancer lacks FOXA binding (**Updated Fig. 2d – was Fig. 2c in original submission**), accessible chromatin (**Supp. Fig. 4b**), and H3K4me1 signal (**Supp. Fig. 4b**) prior to pancreas induction. Furthermore, in the absence of PDX1, FOXA does not bind to the enhancer and enhancer chromatin does not gain accessibility or activation marks (**Fig. 5c**). Consistent with our hypothesis that FOXA motif strength determines temporal FOXA binding, optimizing FOXA motifs resulted in FOXA recruitment to the *NKX6.1* enhancer prior to pancreatic lineage induction. This early recruitment of FOXA led to deposition of H3K4me1 prior to enhancer activation, showing that FOXA mediates H3K4me1 deposition without activating the enhancer. Thus, we show that optimization

of FOXA binding motifs can uncouple FOXA recruitment from cooperative interactions with PDX1 and convert an unprimed enhancer into a primed enhancer.

Given potential cell heterogeneity at the PP2 stage, we further conducted single cell RNA-seq analysis of PP2 stage cells from control and motif optimized cell lines to determine how FOXA motif strength affects gene expression and PP2 stage cell type composition. While *NKX6.1* expression in control PP2 stage cells was restricted to a subpopulation of cells expressing high levels of *PDX1*, *NKX6.1* was more broadly expressed in motif optimized cultures, including in cells expressing lower levels of *PDX1*. These findings indicate that optimizing FOXA motif strength lowers the threshold for enhancer activation and *NKX6.1* target gene expression, which we confirmed at the protein level. Thus, we show that increasing FOXA motif strength renders *NKX6.1* expression independent of high levels of PDX1, which corroborates our model and conclusions. Given that alpha cells are derived from *NKX6.1*⁻ endocrine progenitors, whereas beta cells arise from *NKX6.1*⁺ endocrine progenitors¹, we further examined effects of ectopic expression of *NKX6.1* on cell fate allocation and observed a pre-alpha to a pre-beta cell fate shift. Therefore, our results provide evidence for the importance of FOXA binding mechanisms at developmental enhancers for cell type specification.

These new data not only validate the model proposed in our initial submission but substantially expand insights by showing that changes in FOXA binding motif strength (i) redefine patterns of FOXA recruitment during development, (ii) uncouple FOXA binding from lineage-determining TFs, (iii) alter the threshold for target gene expression, and (iv) change cell fate choice. While other studies have reported a correlation between FOXA motif strength and the ability of FOXA to displace nucleosomes²⁻⁴, our results are the first to directly demonstrate that motif strength is a determinant of FOXA binding affinity *in vivo* and that motif strength helps fine-tune developmental gene regulation with relevance to organ cell type composition.

Specific points

1. P 5-6 L 111 What about DE or GT stage-what is the evidence that the double KO cells are equivalent to wild type? Expression of two markers does not prove this. The WT and KO are somewhat separated in PC1 in Figure 1g and clearly in PC2 in Supp. 2C.

We agree that some degree of dysregulation in gene expression and enhancer activation exists between control and *FOXA1/2* mutants prior to pancreas induction. We edited the Results section to point out this difference and made the following edits:

L128-130: "*Principle component analysis (PCA) of transcriptome data further confirmed that FOXA1/2^{-/-} and control cells were more similar at the GT stage than at the PP2 stage (Fig. 1g).*"

L144-145: “Like gene expression (**Fig. 1g**), H3K27ac profiles in FOXA1/2^{-/-} and control cells differed more substantially at the PP2 than at the GT stage (**Supplementary Fig. 3c**),...”

2. L 156. What is the role of Class 3 Enhancers-what are they doing at GT stage? Just because they are a minority does not mean they do not have significant function. Also they seem to have higher H3K27ac at PP2 in Figure 2b.

To determine a potential role for class III enhancers, we quantified H3K27ac, FOXA1, and FOXA2 ChIP-seq signal at these enhancers during pancreas induction. As shown in **Reviewer 3 Figure 1a-c**, we find similar patterns of H3K27ac signal between class I, II, and III enhancers. Examination of FOXA1 and FOXA2 signal revealed that class III enhancers exhibit relatively low FOXA1 signal at both the GT and PP2 stages, while FOXA2 signal is higher at the GT stage than the PP2 stage, indicating that FOXA2 may be evicted from class III enhancers during pancreas induction. To determine the biological relevance of class III enhancers, we examined activity of the enhancers in FOXA1/2 mutants. We found that activation of class III enhancers (i.e., H3K27ac deposition) depends on FOXA TFs. We added these data as **Updated Supp. Fig. 3g**. One potential mechanism at this class of enhancers is that FOXA TFs could displace histones prior to pancreas induction and then be replaced by other TFs during pancreas induction. A similar mechanism has been shown for Zelda TFs in *Drosophila*¹⁶. However, we find little overlap with binding sites for pancreatic TFs at the PP2 stage (6%, 7%, and 5% of class III enhancers bind PDX1, SOX9, or HNF6, respectively), and differential motif enrichment analysis of class III enhancers yields no compelling candidates. Therefore, the identity of potential TFs to replace FOXA TFs upon pancreas induction remains unclear.

We also found that at several class III enhancers FOXA binding is reduced at the PP2 stage, but not entirely lost (**Reviewer Figure 1d**). It is therefore possible that many of the class III enhancers represent weak FOXA binding sites which are just above the threshold of peak calling at the GT stage but just below this threshold at the PP2 stage. Consistent with this possibility, only 14% of class III enhancers show a significant loss of FOXA ChIP-seq signal (≥ 2 -fold decrease, $P \text{ adj.} < 0.05$) between the GT and PP2 stage. Thus, many class III enhancers may be assigned to this category due to peak calling thresholds, which would indicate that they are a technical artifact rather than presenting distinct biology. In other words, class III enhancers may represent a subset of class I enhancers where FOXA binding signal is lower in PP2 than in GT.

We updated our results section to show class III enhancers' dependency on FOXA TFs for activation (**Updated Supp. Fig. 3g**).

L167-172: “Analysis of H3K27ac signal intensity at the GT and PP2 stages showed similar patterns of H3K27ac signal at all enhancers (**Fig. 2b**), suggesting that enhancers of all classes are mostly inactive at the GT stage and become activated during pancreatic lineage induction. Activation of enhancers of all classes during the GT to PP2 transition was dependent on FOXA1/2 (**Fig. 2c and Supplementary Fig. 3g**).”

Given that the assignment to this class has likely technical reasons, we believe that our analysis is better focused on only class I and class II enhancers. Upon the reviewer's request we will include the data shown in the reviewer figure into the manuscript.

Reviewer 3 Figure 1. Class III pancreatic enhancers gain H3K27ac signal but lose FOXA2 signal during pancreas induction. (a) Box plot of H3K27ac ChIP-seq counts at class I, class II, and class III pancreatic enhancers at GT and PP2 stages ($P = < 2.2 \times 10^{-16}$, $< 2.2 \times 10^{-16}$, and $< 2.2 \times 10^{-16}$ for comparisons of GT versus PP2 stages at class I, class II, and class III enhancers, respectively; Wilcoxon rank sum test, 2-sided). (b) Box plot of FOXA1 ChIP-seq counts at class I, class II, and class III pancreatic enhancers in GT and PP2 stages ($P = < 2.2 \times 10^{-16}$, $< 2.2 \times 10^{-16}$, and 0.85 for comparisons of GT versus PP2 stages at class I, class II, and class III enhancers, respectively; Wilcoxon rank sum test, 2-sided). (c) Box plot of FOXA2 ChIP-seq counts at class I, class II, and class III pancreatic enhancers at GT and PP2 stages ($P = 6.06 \times 10^{-13}$, $< 2.2 \times 10^{-16}$, and $< 5.96 \times 10^{-10}$ for comparisons of GT versus PP2 stages at class I, class II, and class III enhancers, respectively; Wilcoxon rank sum test, 2-sided). All ChIP-seq experiments, $n = 2$ replicates from independent differentiations. (d) Genome browser snapshot showing FOXA1, FOXA2, and H3K27ac ChIP-seq signal in control and *FOXA1/2^{-/-}* GT and PP2 stage cells at class III pancreatic enhancer near *HNF1B*.

3. L 159 is there a “bookmarking” function of Class 1 enhancers?

This is an intriguing possibility. FOXA1 has been shown to bind to a subset of liver differentiation genes during mitosis in developing cells¹⁷. However, this prior study identified strong FOXA motifs at both bookmarked and non-bookmarked FOXA1 binding sites, which is inconsistent with our observations at class I compared to class II enhancers. Given these discrepancies and the need for biochemical validation to confirm this possibility, we believe this question is better left for future studies.

4. L 188 how are superenhancers defined/identified here

To define pancreatic super-enhancers, we first identified pancreatic enhancers as distal genomic regions exhibiting a ≥ 2 -fold increase in H3K27ac ChIP-seq signal during pancreas induction. We then used Rank Ordering of Super-enhancers (ROSE) software^{14,15} to join identified pancreatic enhancers within a 12.5 kb span and rank these joined enhancers based on intensity of H3K27ac ChIP-seq signal. These joined enhancers were plotted based on H3K27ac signal as in **Supp. Fig. 4d**, and pancreatic super-enhancers were defined as joined enhancers ranking above the inflection point of the resulting graph. We apologize that this was unclear in our initial submission and have updated the Methods section accordingly (L967-975).

5. Fig 3e more in depth analysis of the class of genes associated with Class 1 and 2 enhancers would be informative

We performed comparative gene ontology analysis of genes associated with class I enhancers compared to those associated with class II enhancers and vice versa (**Updated Supp. Fig. 4g**). Interestingly, genes associated with class I enhancers showed enrichment for GO terms related to cell signaling processes, suggesting that genes regulated by class I enhancers modulate intracellular signaling events in lineage intermediates. This finding, combined with the enrichment for binding motifs of signal-dependent TFs (**Updated Fig. 4a**) and preferential binding of the retinoic receptor subunit RXR at class I enhancers (**Updated Fig. 4b**), provide evidence that environmental signals are interpreted at class I enhancers. We found no comparatively enriched GO categories for genes associated with class II enhancers.

6. L216 Fig 4a are these the only enriched motifs? What determined the cutoff for display in this figure?

Motifs of relevant transcription factors were chosen for display from a list generated by HOMER. For reference, the full list of all enriched motifs across comparisons of class I and class II pancreatic enhancers was included in the initial submission as **Supplementary Table 4** (also **Supplementary Table 4** in revised submission).

7. L 233 4d explain terminology here for those unfamiliar with this field, what do MAO refer to? How were these singled out for analysis?

The MA numbers are unique identifiers assigned to each binding motif position weighted matrix (PWM) within the JASPAR motif database: <http://jaspar.genereg.net/docs/>. These PWMs are derived from published collections of experimentally defined TF binding sites, where the size of each letter in the matrix corresponds to its relative importance in determining binding of the TF^{18,19}.

For our analysis, we selected the three most divergent PWMs for FOXA1 and FOXA2 from the JASPAR database. A similar strategy was used in a recent analysis of pioneer factor binding sites². We updated the Methods section to include a more thorough explanation.

L1000-1001: “FOXA1 and FOXA2 PWMs were selected to encompass the most divergent PWMs for each TF. PWMs were downloaded from the JASPAR database.”

8. L 316 and Figure 6d FOXA1 signal increased at class1 in ALV stage, apparent contrast to pancreas and liver?

FOXA signal increases during lineage induction at class I enhancers across endodermal lineages, as shown below in **Reviewer 3 Figure 2**.

Reviewer 3 Figure 2. Class I pancreatic, hepatic, and alveolar enhancers gain FOXA signal upon lineage induction. (a) Box plot of FOXA1 and FOXA2 ChIP-seq counts at class I pancreatic enhancers in GT and PP2 stages ($P = < 2.2 \times 10^{-16}$ and 6.06×10^{-13} for comparisons of FOXA1 and FOXA2 ChIP-seq signal, respectively; Wilcoxon rank sum test, 2-sided). (b) Box plot of FOXA1 and FOXA2 ChIP-seq counts at class I hepatic enhancers in GT and HP stages ($P = 1.24 \times 10^{-15}$ and $< 2.2 \times 10^{-16}$ for comparisons of FOXA1 and FOXA2 ChIP-seq signal, respectively; Wilcoxon rank sum test, 2-sided). (c) Box plot of FOXA1 ChIP-seq counts at class I alveolar enhancers in AFG and ALV stages ($P < 2.2 \times 10^{-16}$; Wilcoxon rank sum test, 2-sided). All ChIP-seq experiments, $n = 2$ replicates from independent differentiations.

9. Figure 6 g,h-similar question to 6 above.

The full lists of all enriched motifs across comparisons of class I and class II liver and lung enhancers were included in the original and revised submission as **Supplementary Table 6**.

10. L 414 function of primed enhancers remains unclear, what do the authors think pioneering activity really means?

11. L 442 onward-this is somewhat confusing and speculative.

Our data indicate that primed enhancers are the ones that interpret environmental signals and are therefore critical for initiating lineage-specific gene expression programs in response to lineage-inductive signals. This conclusion is supported by (i) motif enrichment for signal-dependent TFs at primed enhancers (**Updated Fig. 4a** and **Fig. 7g,h**), (ii) binding of the retinoic acid receptor subunit RXR to primed enhancers (**Updated Fig. 4b**), and (iii) enrichment for components of signal transduction pathways among genes associated with primed enhancers (**Updated Supp. Fig. 4g**).

Furthermore, we experimentally demonstrate that converting an unprimed to a primed enhancer broadens the domain of target gene expression within the pancreatic

progenitor population (**Updated Fig. 6c**), suggesting that primed enhancers allow for more promiscuous gene expression across organ progenitors.

Based on these findings we propose that primed enhancers initiate organ lineage programs by responding to lineage inductive cues and broadly inducing the expression of the first lineage determining TFs in organ progenitors. In support of this, we found primed enhancers near *PDX1*, *HNF1B*, and *MEIS1*, which are among the first TFs expressed upon pancreas induction. We propose that the higher thresholds to target gene expression conferred by unprimed enhancers help restrict gene expression to specific cell populations, as enhancer activation will only occur when specific complements of lineage-specific TFs are present in sufficient concentrations. Thus, gene regulation by unprimed enhancers provides a mechanism for specifying different cell types in organ development.

We revised the entire Discussion section to better convey the specific roles of primed and unprimed enhancers in early endodermal organ development.

(L509-551): *“Our findings provide insight into the gene regulatory mechanisms that underlie endodermal organ lineage induction and cell fate specification. We observed enrichment of binding motifs for signal-dependent TFs and binding of the retinoic acid receptor subunit RXR at primed pancreatic enhancers. These findings suggest that organ lineage-inductive cues are read by primed enhancers to initiate expression of lineage-determining TFs. In support of this, primed enhancers are found near PDX1, HNF1B, and MEIS1, which are among the first TFs expressed upon pancreas induction. By contrast, unprimed enhancers are enriched for binding motifs of organ-specific TFs which recruit FOXA1/2 secondarily. Given that FOXA TFs are broadly expressed across endodermal organ lineages, indirect FOXA recruitment by organ-specific TFs provides a safeguard against lineage-aberrant enhancer activation and gene expression. This agrees with studies in Drosophila and Ciona which suggest that suboptimization of TF binding motifs could be a general principle by which to confer cell specificity to enhancers^{26,39}.*

Replacing low affinity FOXA binding sites at an unprimed enhancer for NKX6.1 with higher affinity sites broadened the domain of NKX6.1 expression among pancreatic progenitors. As we show, NKX6.1 was not prematurely expressed, demonstrating that motif optimization does not eliminate the dependency of target gene expression on lineage-specific cues. This suggests that early FOXA recruitment through high affinity binding sites lowers the threshold for target gene expression, which could reflect an increased sensitivity of the enhancer to activation by lineage-specific TFs. Thus, higher thresholds to target gene expression conferred by unprimed enhancers will restrict target gene expression to specific cell populations, as enhancer activation will only occur when a specific complement of lineage-specific TFs is present in sufficient concentrations. We propose that gene regulation by unprimed enhancers provides a mechanism for specifying different cell types early in organ development. Small differences in TF expression among early organ progenitors would be sufficient to activate different repertoires of unprimed enhancers, thereby creating divergent gene expression patterns and cell populations. Consistent with this concept, it has been

shown that PDX1^{high} and PDX1^{low} cells in the early pancreatic epithelium acquire different cell identities⁴⁰.

We demonstrate that conversion of a single enhancer near NKX6.1 from an unprimed to a primed state is sufficient to alter cell fate due to broadened expression of NKX6.1 within the progenitor cell domain. These findings show that in a developmental context, differences in FOXA binding affinity at enhancers can affect cell fate allocation. It is therefore possible that polymorphisms at FOXA binding sites determine interindividual differences in endodermal organ cell type composition. Consistent with this possibility, islet cell type composition is known to vary greatly in humans⁴¹ and the NKX6.1 enhancer contains twelve known polymorphisms predicted to alter the strength and spacing of FOXA motifs. While the importance of polymorphisms for organ cell type composition remains to be demonstrated, our findings support the concept that FOXA TF motif strength at developmental enhancers provides a tunable threshold for target gene expression.”

12. L 457 why only a subset of primed hepatic enhancers lost after pancreas induction. What is this subset?

We characterized FOXA TF binding to hepatic enhancers immediately after pancreas induction. We speculate that at this time point the “eviction” of FOXA from enhancers of the alternative lineage has only been initiated, thus explaining the small subset. It has been shown that pancreatic progenitors can activate liver-specific genes, albeit only during a short competence window shortly after pancreas induction²⁰. This early competence to express liver genes might be functionally related to incomplete FOXA eviction from hepatic enhancers early in pancreas development. Further investigation of this hypothesis will require additional analyses which we believe are beyond the scope of this study. Since the “eviction” of FOXA is a minor point of our study, we removed the section from the Discussion.

References

- 1 Schaffer, A. E. *et al.* Nkx6.1 controls a gene regulatory network required for establishing and maintaining pancreatic Beta cell identity. *PLoS Genet* **9**, e1003274, doi:10.1371/journal.pgen.1003274 (2013).
- 2 Donaghey, J. *et al.* Genetic determinants and epigenetic effects of pioneer-factor occupancy. *Nat Genet* **50**, 250-258, doi:10.1038/s41588-017-0034-3 (2018).
- 3 Swinstead, E. E. *et al.* Steroid Receptors Reprogram FoxA1 Occupancy through Dynamic Chromatin Transitions. *Cell* **165**, 593-605, doi:10.1016/j.cell.2016.02.067 (2016).
- 4 Zhang, Y. *et al.* GENE REGULATION. Discrete functions of nuclear receptor Rev-erbalpha couple metabolism to the clock. *Science* **348**, 1488-1492, doi:10.1126/science.aab3021 (2015).
- 5 Rada-Iglesias, A. *et al.* A unique chromatin signature uncovers early developmental enhancers in humans. *Nature* **470**, 279-283, doi:10.1038/nature09692 (2011).
- 6 Calo, E. & Wysocka, J. Modification of enhancer chromatin: what, how, and why? *Mol Cell* **49**, 825-837, doi:10.1016/j.molcel.2013.01.038 (2013).
- 7 Jozwik, K. M., Chernukhin, I., Serandour, A. A., Nagarajan, S. & Carroll, J. S. FOXA1 Directs H3K4 Monomethylation at Enhancers via Recruitment of the Methyltransferase MLL3. *Cell Rep* **17**, 2715-2723, doi:10.1016/j.celrep.2016.11.028 (2016).
- 8 Gao, S. *et al.* Chromatin binding of FOXA1 is promoted by LSD1-mediated demethylation in prostate cancer. *Nat Genet* **52**, 1011-1017, doi:10.1038/s41588-020-0681-7 (2020).
- 9 Korkmaz, G. *et al.* A CRISPR-Cas9 screen identifies essential CTCF anchor sites for estrogen receptor-driven breast cancer cell proliferation. *Nucleic Acids Res* **47**, 9557-9572, doi:10.1093/nar/gkz675 (2019).
- 10 Rojas, D. A. *et al.* Increase in secreted airway mucins and partial Muc5b STAT6/FoxA2 regulation during Pneumocystis primary infection. *Sci Rep* **9**, 2078, doi:10.1038/s41598-019-39079-4 (2019).
- 11 Pasquali, L. *et al.* Pancreatic islet enhancer clusters enriched in type 2 diabetes risk-associated variants. *Nat Genet* **46**, 136-143, doi:10.1038/ng.2870 (2014).
- 12 Weedon, M. N. *et al.* Recessive mutations in a distal PTF1A enhancer cause isolated pancreatic agenesis. *Nat Genet* **46**, 61-64, doi:10.1038/ng.2826 (2014).
- 13 Lee, K. *et al.* FOXA2 Is Required for Enhancer Priming during Pancreatic Differentiation. *Cell Rep* **28**, 382-393 e387, doi:10.1016/j.celrep.2019.06.034 (2019).
- 14 Whyte, W. A. *et al.* Master transcription factors and mediator establish super-enhancers at key cell identity genes. *Cell* **153**, 307-319, doi:10.1016/j.cell.2013.03.035 (2013).
- 15 Loven, J. *et al.* Selective inhibition of tumor oncogenes by disruption of super-enhancers. *Cell* **153**, 320-334, doi:10.1016/j.cell.2013.03.036 (2013).
- 16 Liang, H. L. *et al.* The zinc-finger protein Zelda is a key activator of the early zygotic genome in Drosophila. *Nature* **456**, 400-403, doi:10.1038/nature07388 (2008).

- 17 Caravaca, J. M. *et al.* Bookmarking by specific and nonspecific binding of FoxA1 pioneer factor to mitotic chromosomes. *Genes Dev* **27**, 251-260, doi:10.1101/gad.206458.112 (2013).
- 18 Stormo, G. D. Modeling the specificity of protein-DNA interactions. *Quant Biol* **1**, 115-130, doi:10.1007/s40484-013-0012-4 (2013).
- 19 Wasserman, W. W. & Sandelin, A. Applied bioinformatics for the identification of regulatory elements. *Nat Rev Genet* **5**, 276-287, doi:10.1038/nrg1315 (2004).
- 20 Seymour, P. A. *et al.* A Sox9/Fgf feed-forward loop maintains pancreatic organ identity. *Development* **139**, 3363-3372, doi:10.1242/dev.078733 (2012).

REVIEWERS' COMMENTS

Reviewer #3 (Remarks to the Author):

In their revision the authors report additional experimental studies to characterize FOXA binding at specific stages and show the importance of the strength of binding to cell fate decisions. They have also revised the discussion to clarify their overall interpretation of the findings and their significance. These revisions have improved the manuscript and addressed some important concerns of the reviewers.